# Interpretable candidate drug prioritization and explanation framework across-medical knowledge graphs based on graph embedding models: A case study of type 2 diabetes

Zekun Zhou, Shuo Yang*

Institute of Information on Traditional Chinese Medicine, China Academy of Chinese Medical Sciences, Beijing, China

* 591007703@qq.com

## Abstract

### Objective

Addressing the challenges in elucidating the mechanisms of complex diseases such as Type 2 Diabetes Mellitus (T2DM), this study aims to construct a domain-specific cross-medicine knowledge graph (CMKG) and develop a unified path scoring framework that couples graph embeddings with rule-based reasoning, enabling high-precision, interpretable prioritization and explanation of potential drug candidates.

### Methods

First, multi-source biomedical data from Hetionet, SymMap, TCMBank, STRING, and TTD were integrated. Using Jaccard and overlap-based fusion strategies, entity alignment and relation consolidation were performed to construct a deep CMKG bridged by genes. Second, four graph embedding models (TransE, DistMult, ComplEx, and RotatE) were introduced for link prediction and evaluated using MRR and Hits@K. Finally, to overcome the interpretability limitations of black-box predictions, AnyBURL rule learning was combined with depth-first search (DFS). We innovatively introduced an Ingredient Specificity Index (ISI) and a hybrid path confidence calibration mechanism, constructing a unified path scoring system incorporating length decay, node/ relation weights, and experimental evidence bonuses to screen the most critical mechanistic paths.

### Results

The constructed CMKG contains 15 entity types (245,235 entities) and 52 relation types (7,155,373 triples), covering 709 core T2DM genes. Link prediction stability tests across multiple random seeds showed that the ComplEx model consistently performed best in handling complex multi-mapping relations (MRR = 0.213 ± 0.004,

**Data availability statement:** Data and source code are available at:https://github.com/kunkunabc/T2DM-CrossMed-KG/tree/main.

**Funding:** The author(s) received no specific funding for this work.

**Competing interests:** The authors have declared that no competing interests exist.

Hits@10 = 0.418 ± 0.003). Consequently, the fully converged ComplEx model (Peak Hits@10 = 0.48) was utilized for comprehensive prediction. Retaining the top 100 predictions, Abelmoschus manihot and Topiramate ranked highest among TCM herbs and modern medicine compounds, respectively. Path analysis based on the scoring system revealed deep multi-target mechanisms, including insulin signaling sensitization, inflammatory regulation, and chromatin/cell-cycle intervention.

## Conclusion

The proposed gene-bridged graph embedding and unified path scoring framework successfully translates probabilistic predictions into biologically traceable semantic explanations. Rigorous ablation and parameter sensitivity experiments confirm that the framework achieves a robust balance between knowledge coverage and explanatory specificity, providing a transparent, robust, and scalable methodological foundation for candidate drug prioritization in complex diseases.

---

## 1. Introduction

Diabetes is a chronic metabolic disease whose global prevalence continues to rise. A 2023 study in *The Lancet* reported 529 million people living with diabetes in 2021, a number projected to exceed 1.31 billion by 2050 [1]. This sustained growth has intensified the burden of disease treatment worldwide and drawn mounting attention to diabetes. In this context, research on drug discovery, which is closely tied to the development of antidiabetic therapeutics, has become a critical focus in medicine. Traditional drug development is time-consuming, costly, and marked by low success rates. According to *JAMA Network Open (2024)*, the average cost to develop a new drug is approximately USD 172.7 million, rising to USD 515.8 million when the cost of failures are included [2]. Consequently, knowledge mining from multi-source biomedical databases has emerged as a promising direction for more efficient and cost-effective drug discovery.

On the policy front, the Healthy China 2030 blueprint calls for advancing the informatization of diabetes management and the sharing of health big data. As a powerful semantic network tool, knowledge graphs can integrate multi-source heterogeneous data and represent biomedical entities and their relations in a structured form, supporting knowledge reasoning and drug discovery [3]. Traditional Chinese Medicine (TCM) and Modern Medicine (MM) differ in nomenclature systems and theoretical underpinnings: TCM emphasizes holism and pattern-based treatment, focusing on internal homeostasis and individual differences, whereas MM grounds diagnosis and therapy in cellular and molecular mechanisms. Using gene targets as a bridge to construct a cross-medicine knowledge graph enables the fusion of both systems, facilitating resource sharing and theoretical complementarity.

Moreover, in the field of cross-medicine drug prediction, although existing integrative knowledge graphs provide a foundation for cross-system associations, they often lack in-depth mechanistic characterizations for specific complex diseases.

Generic graph construction tends to overlook high-dimensional features and heterogeneous interactions within a specific disease context, making it difficult to directly support deep mechanistic reasoning. As pointed out by Li et al. [4], constructing domain-specific integrated knowledge graphs (such as for rheumatology) is crucial for mining implicit knowledge from ancient literature. Similarly, Li and Hu et al. [5] explored the data privacy and ethical challenges in integrating AI with TCM, suggesting that precision and contextual adaptability must be prioritized during graph construction. By leveraging domain-specific knowledge graphs in combination with graph embedding models, the complex relations within the graph can be mapped into low-dimensional vectors, enabling the prediction of potential drug-disease and drug-gene associations. However, the inherent "black-box" nature of these models often results in a lack of interpretability in their predictions. To address this issue, this study focuses on Type 2 Diabetes Mellitus (T2DM), a major metabolic disease, to construct a domain-specific Cross-Medical Knowledge Graph (CMKG). We further integrate two interpretable methods—rule learning and depth-first search (DFS)—to establish a unified path scoring mechanism. By allocating differentiated weights to various factors along the path based on biomedical and molecular mechanism characteristics, this approach can efficiently screen the most critical action pathways for the predicted drugs, effectively resolving the issue of poor interpretability. Specifically, using TCM and modern medicine data from SymMap and TCMBank as the backbone, we directionally fused systems biology data from Hetionet, molecular interaction networks from STRING, and clinical validation data from TTD, tailored to the pathological features of T2DM. This "scenario-driven" construction strategy aims to overcome the insufficient mechanistic resolution of generic graphs when applied to specific diseases, thereby providing a high-precision, interpretable data foundation for multi-target candidate drug prioritization and explanation in diseases like T2DM.

Table 1 summarizes the statement of significance of this work in terms of the problem addressed, what is already known, and what this paper adds. In summary, this paper makes three main contributions: First, we reconstruct a large-scale, gene-bridged CMKG for T2DM that systematically integrates multi-source biomedical data from Hetionet, SymMap, TCMBank, STRING, and TTD, effectively filling the gap in molecular pathway connectivity prevalent in traditional TCM databases. Second, we evaluate and compare four representative graph embedding models (TransE, DistMult, ComplEx, and RotatE) for link prediction on this large-scale heterogeneous cross-medicine graph. We utilized the multi-seed tests as rigorous statistical validation to establish model stability and relative ranking, ultimately selecting the complex space model (ComplEx) that performs best on complex multi-mapping relations for comprehensive candidate drug prediction. Third, we propose an innovative unified interpretable reasoning framework that couples AnyBURL symbolic rule learning with DFS-based graph search. By introducing an Ingredient Specificity Index (ISI) and a hybrid path confidence calibration mechanism, this framework effectively breaks the limitations of "black-box" predictions, generating high-confidence, biologically specific, and interpretable mechanistic pathways for the predicted candidate therapeutics (such as Abelmoschus manihot and topiramate).

The rest of this paper is organized as follows. In Section 2, we review related work on current methods for new drug research and development, interpretable reasoning over knowledge graphs, and diabetes-related knowledge graphs across TCM and MM. Section 3 describes the materials and methods, including the overall workflow, data collection and preprocessing, and the construction and analysis of the CMKG via entity alignment and relation fusion. Section 4 details the methodology implementation, covering the graph embedding models for link prediction, experimental settings, and the development of the unified path scoring framework incorporating the ISI and confidence calibration mechanism. Section 5 presents the results and discussion, encompassing link prediction performance, prediction of candidate therapeutics, parameter sensitivity analysis, ablation studies, and interpretability analysis of key semantic paths. Finally, Section 6 concludes the paper and discusses potential limitations and directions for future research.

## 2. Related work

### 2.1 Current research on methods for new drug R&D

Recent advances in science and technology have driven profound innovation across the drug development pipeline, spanning molecular design, drug delivery, and clinical trials. Artificial intelligence, particularly machine learning and

**Table 1. Statement of significance.**

| Summary | Description |
|---|---|
| Problem | Drug discovery for Type 2 Diabetes Mellitus (T2DM) remains inefficient, and knowledge from Traditional Chinese Medicine (TCM) and Modern Medicine (MM) is fragmented across heterogeneous databases. Existing graph embedding-based prediction methods typically function as "black boxes," lacking interpretable mechanistic explanations with sufficient resolution for specific complex diseases. |
| What is already known | Biomedical knowledge graphs and embedding models have been applied for drug-disease and drug-gene prediction, and rule-based approaches offer partial interpretability. However, prior studies generally focus on a single medical system, fail to construct high-resolution networks tailored to specific disease contexts, and rarely couple graph embeddings with explicit, biologically specific path scoring within a unified computational framework. |
| What this paper adds | This paper constructs a domain-specific, gene-bridged Cross-Medical Knowledge Graph (CMKG) for T2DM. It develops a unified interpretability framework integrating a complex space embedding model (ComplEx), AnyBURL rule learning, and a sophisticated path scoring system (incorporating an Ingredient Specificity Index and hybrid confidence calibration). This approach identifies biologically specific candidate therapeutics from both TCM and MM, revealing multi-target mechanisms and providing a transparent, robust, and scalable methodology for candidate drug prioritization and explanation in complex diseases. |

deep learning, plays a vital role. For example, Chen Haoyu [6] combined generative adversarial networks with evolutionary algorithms to construct a generative model for novel drug molecules, improving molecular quality and data diversity, and used a semi-supervised interaction network to effectively predict potential drug-disease interactions. Significant progress has also been made in novel drug delivery systems. Nanotechnology has delivered notable results: lipid nanoparticles (LNPs) [7] have achieved great success in delivering mRNA therapeutics (e.g., COVID-19 vaccines), demonstrating potential for enhancing efficacy and reducing adverse effects. For instance, the Anderson team [8] used high-throughput screening to synthesize large numbers of ionizable lipids and developed efficient inhalable carriers for gene editing in the pulmonary epithelium; the Siegwart team [9] designed multi-tailed ionizable phospholipids (iPhos) that promote membrane conversion and mRNA release in acidic environments, addressing low endosomal escape efficiency and opening new avenues for gene therapy and vaccine development. Drug design based on Structural Biology has likewise advanced. In May 2024, Google Deep Mind released AlphaFold3 [10], which uses diffusion models to substantially improve the accuracy of predicting structures of protein-ligand and nucleic acid complexes. Although AI excels in single-protein structure prediction, analyzing multi-protein complexes and dynamic structures still calls for integration of experimental techniques with computational models; high-resolution experimental structures remain crucial for future breakthroughs. In clinical development, adaptive trial designs are seeing broader adoption. These designs allow pre-specified modifications during a trial, such as adjusting sample size, changing dose, or revising eligibility criteria, thereby improving efficiency and success rates [11]. Moreover, the use of real-world evidence in new drug R&D is drawing more attention. By analyzing health e-records and medical insurance data, researchers can better understand a drug's effectiveness and safety in routine clinical practice [12]. In drug discovery based on knowledge graph, promising results have already emerged. For example, Wu Jiamin [13] built a lung cancer medical knowledge graph and used algorithms to help uncover treatment pathways; Nordon et al. [14] constructed concept graphs from medical literature and electronic health records to mine causal links between diseases and drugs, identifying candidate drug families for hypertension and diabetes repurposing. By integrating multi-source heterogeneous data, knowledge graphs provide structured support for analyzing disease–drug–gene associations, improving the efficiency of drug discovery and making the R&D process more scientific and interpretable. Other studies have shown that fine-grained network structural information is highly valuable for predicting associations between molecules or biological entities. For instance, the MotifMDA model proposed by Zhao et al. [15] significantly improved the performance of miRNA-disease

association prediction by designing several motifs that characterize these associations and employing a two-layer hierarchical attention network to couple high-order and low-order structural information. This suggests that incorporating motif-level structural features into network analyses for drug discovery could further enhance prediction accuracy and interpretability.

Overall, knowledge graphs offer clear advantages in integrating diverse data, pinpointing targets, and predicting drug effects. Constructing disease-drug knowledge graphs can accelerate discovery, optimize development workflows, and provide essential technical support for precision medicine and innovative therapeutics.

## 2.2 Interpretable reasoning over knowledge graphs

Within the connectionist paradigm, graph embedding models aim to map entities and relations in a knowledge graph into a low-dimensional vector space while preserving structural and semantic information, facilitating machine learning and reasoning tasks. Classic translational distance models such as TransE [16] use vector translation to measure relations between entities. Although simple and effective, TransE can suffer semantic conflicts under multi-mapping relations. To address this, Sun et al. [17] proposed RotatE, which represents entities as complex vectors and defines relations as rotations in the complex plane, effectively modeling symmetric and anti-symmetric relations and achieving strong performance in Knowledge Graph Completion Task. Subsequently, Trouillon et al. [18] introduced ComplEx, which also operates in a complex vector space to match entities and relations semantically, handling many-to-many relations and achieving high accuracy in link prediction.

In biomedical knowledge representation, scholars at Yale University [19] applied TransE, ComplEx, and RotatE to the SNOMED-CT knowledge graph (≈2 million triples, 290,000 entities, and 170 relations) in *ACL-BioNLP*. The results showed that ComplEx and RotatE performed best on link prediction; ComplEx achieved superior Hits@10 and MRR, indicating stable performance and strong semantic discrimination in large-scale biomedical graphs. Nevertheless, despite their ability to capture complex relational structure, these models still face interpretability limitations in interdisciplinary graphs, such as cross TCM and MM graphs. Their explanations typically remain at the level of relation patterns and logical forms and lack semantic interpretation of specific predictions, an evident constraint for high-reliability tasks such as biomedical drug prediction.

To enhance interpretability of models, rule learning has become another important line of research. These methods automatically induce logical rules from knowledge graphs to guide reasoning and completion. AnyBURL [20], one of the more efficient tools, uses random path sampling and Horn rule generalization to mine high-quality rules from large graphs, effectively discovering potential entities and relations. Compared with traditional rule learning approaches, AnyBURL offers greater efficiency and scalability, making it suitable for large biomedical graphs. Combining graph embedding with rule-based reasoning can further improve accuracy of prediction and interpretability; however, joint application on large-scale graph data remains relatively rare.

## 2.3 Knowledge graphs for diabetes across TCM and MM

Research on diabetes knowledge graphs has progressed in both TCM and MM domains. For instance, some studies have built integrated TCM and MM graphs for diabetic nephropathy by combining clinical guidelines with real-world clinical data, revealing patterns of coexisting syndromes and shared therapeutic methods [21]. Chai Jiaqi et al. [22] constructed a knowledge graph for TCM treatment of diabetic peripheral neuropathy (DPN) based on literature data and a Neo4j database, defining 12 node types and 4 relation types to support TCM knowledge retrieval and discovery. Zheng Qunli et al. [23] developed an adjuvant diagnosis system for diabetes (TH-SAC, KGDCP models) using literature data to facilitate risk analysis and improve prediction and diagnosis of disease. Shao Minhui et al. [24] built a blood-glucose management graph for people with diabetes by extracting semantic entities from clinical guidelines and related sources and

organizing their relations to aid glycemic control. Liu Yong et al. [25] employed natural language processing to harmonize medical semantics and constructed a diabetes medical knowledge graph to enhance patients' own awareness and self-management. Using real-world clinical data, Wang Yifei [26] mined TCM diagnostic and therapeutic knowledge for diabetic nephropathy, creating a graph with 903 nodes and 1,670 relations. This work systematically organized TCM knowledge for diabetic kidney disease, supporting the formulation and optimization of clinical regimens and laying a foundation for intelligent treatment recommendation systems.

Overall, T2DM drug discovery based on knowledge graphs faces a dual challenge: first, data heterogeneity across medical systems hinders a unified mechanistic understanding; second, deep learning-based predictive models lack interpretability. Although graph neural networks (GNNs) have made significant progress in areas like drug-drug interaction (DDI) prediction—such as the multi-view contrastive learning framework proposed by Li et al. [27] which effectively improved prediction accuracy, and the work by Li et al. [28] which leveraged Large Language Models (LLMs) to enhance molecular representation learning —these methods often struggle to provide intuitive, path-level explanations when processing complex semantics across traditional and modern medicine. Existing end-to-end models, despite their superior performance, act as "black boxes," making it difficult for clinicians or researchers to understand the specific molecular cascades responsible for a drug's efficacy. Therefore, there is a critical need for an interpretable framework capable of integrating multi-source heterogeneous knowledge while explicitly outputting reasoning paths. To address this challenge, this study proposes a hybrid framework that couples graph embeddings with rule-based reasoning. This approach aims to break the limitations of "black-box" predictions and provide transparent, traceable mechanistic explanations for T2DM candidate drug prioritization.

## 3. Materials and methods

The interpretable candidate drug prioritization framework proposed in this study aims to mine potential therapeutics for T2DM and their underlying molecular mechanisms by integrating multi-source heterogeneous data across TCM and MM. As illustrated in Fig. 1, the overall workflow comprises four sequential stages. First, in the data integration phase, TCM and MM entities and relations were systematically collected from databases including SymMap, Hetionet, and STRING, followed by rigorous data cleaning and standardization. Subsequently, utilizing genes as the semantic hub connecting both medical systems, a CMKG was constructed through entity alignment and relation fusion. Following graph construction, a complex space embedding model (ComplEx) was employed for link prediction to learn the low-dimensional representations of the graph, calculate potential association probabilities between drugs and T2DM, and screen for candidate therapeutics. Finally, to achieve interpretable reasoning—the core innovation of this framework—we designed a Unified Path Scoring System that couples symbolic rule learning (AnyBURL) with a DFS algorithm. By introducing an ISI and a confidence calibration mechanism, this system accurately identifies high-confidence, biologically specific mechanistic paths from a massive pool of candidates, thereby providing transparent semantic explanations for the otherwise "black-box" prediction results.

### 3.1 Data collection and preprocessing

We comprehensively collected data from five large, publicly available biomedical databases, such as Hetionet, SymMap, TCMBank, STRING, and TTD. From STRING, we retained only human-derived gene interactions with confidence ≥ 0.4 (medium confidence), obtaining 10 semantic relation types to serve as interaction evidence at the gene level. We then integrated TCM and MM data from different sources by entity type and relation type to obtain the required triples. Details of entity types by source and of triple relation types are presented in Table 2 and Table 3, respectively: in total, 15 entity types and 245,235 entities; 32 triple types and 7,342,723 triples.

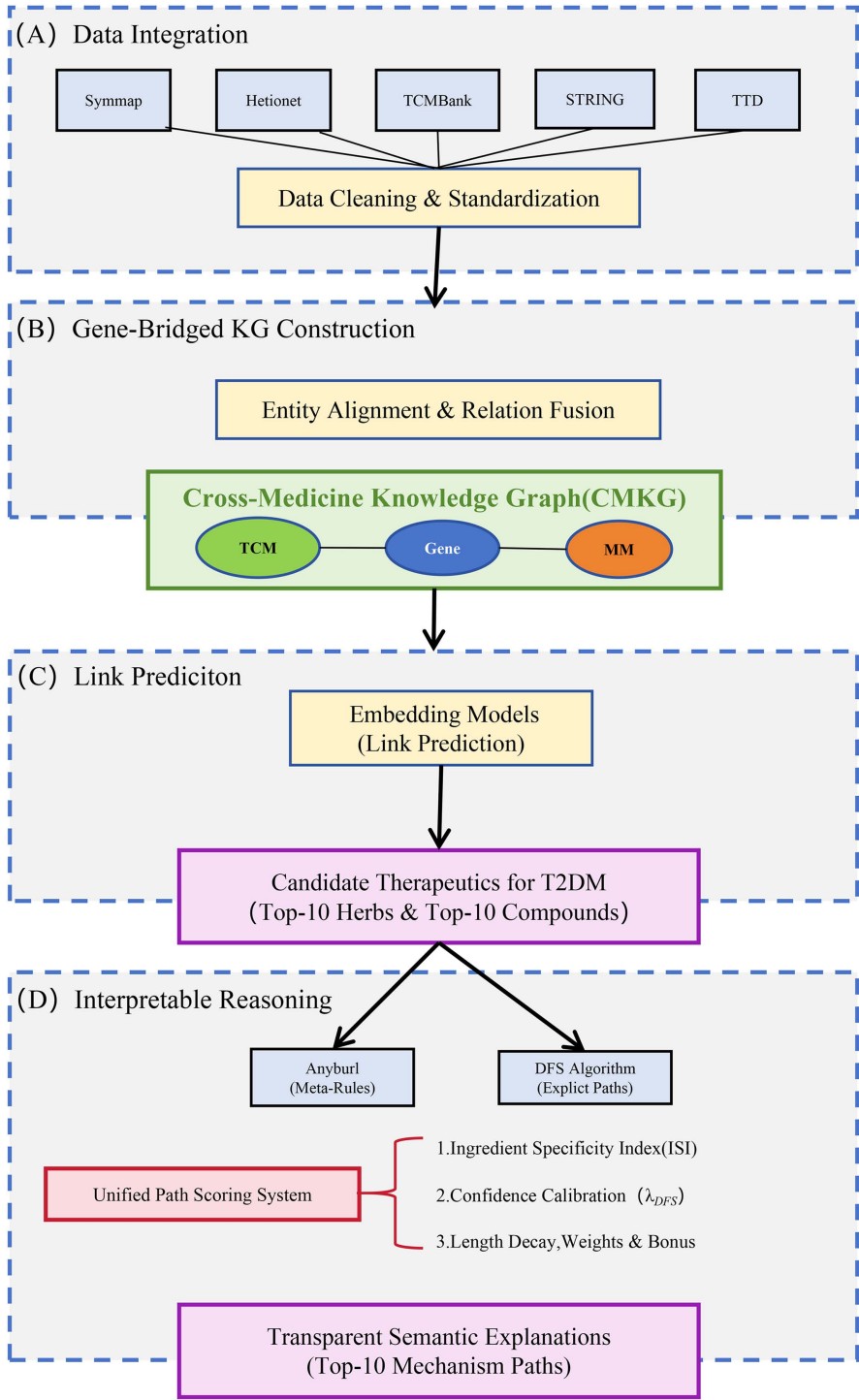

**Fig 1. Overall workflow of the interpretable cross-medicine candidate drug prioritization framework.**

**Table 2. Detailed list of entity types.**

| NO. | Entity Type | Hetionet | Symmap | TCMbank | STRING | TTD | Total |
|---|---|---|---|---|---|---|---|
| 01 | DiseaseMM | 136 | 14426 | 29499 | – | 2343 | 46404 |
| 02 | SymptomMM | 415 | – | 1143 | – | – | 1558 |
| 03 | SymptomTCM | – | 2311 | – | – | – | 3035 |
| 04 | Pharmacological Classification | 345 | – | – | – | – | 345 |
| 05 | Compound | 1538 | – | – | – | 23415 | 24953 |
| 06 | Herb | – | 698 | 6733 | – | – | 8031 |
| 07 | Ingredient | – | 26026 | 35931 | – | – | 61957 |
| 08 | Gene | 19144 | 19028 | 12320 | 19425 | 2613 | 72530 |
| 09 | Pathway | 1822 | – | – | – | 379 | 2201 |
| 10 | Biological Process | 11381 | – | – | – | – | 11381 |
| 11 | Cellular Component | 1391 | – | – | – | – | 1391 |
| 12 | Molecular Function | 2884 | – | – | – | – | 2884 |
| 13 | Anatomy | 400 | – | – | – | – | 400 |
| 14 | Side Effect | 5701 | – | – | – | – | 5701 |
| 15 | Syndrome | – | 233 | – | – | – | 233 |
| 16 | Total | 45157 | 62722 | 85626 | 19425 | 28750 | 245235 |

## 3.2 Knowledge graph construction

**3.2.1 Entity alignment.** Due to differences in entity references and source IDs across databases, the resulting datasets may contain excessive noise and inconsistent mappings, which in turn can adversely affect the model's performance. After deduplicating and removing invalid records, we reassigned custom IDs to entities with inconsistent identifier mappings based on entity names to ensure global uniqueness. The final entity counts and ID naming rules are summarized in Table 4. Among them, "DiseaseMM" comprises 38,346 types, including 75 refined disease entities related to diabetes and its complications; we assigned custom IDs DM01–DM75 (e.g., the focal disease in our case study, "Type 2 Diabetes Mellitus" is DM06). In addition, the symptom entity type was further subdivided according to the two medical systems—Traditional Chinese Medicine (TCM) and Modern Medicine (MM). For example, the entity type "Symptom" was refined into "SymptomTCM" and "SymptomMM", corresponding to the traditional and modern medical systems, respectively.

**3.2.2 Relation fusion.** Given the differing semantic granularities of triple relations across sources, redundancy can remain high. Following prior work [26], we fused relations using the Jaccard Similarity Coefficient in combination with the Overlap Coefficient. The Jaccard Coefficient is given in Equation (1) and the Overlap Coefficient in Equation (2).

$$J(A, B) = \frac{|A \cap B|}{|A \cup B|}$$

(1)

$$O(A, B) = \frac{|A \cap B|}{min\,(|A|, |B|)}$$

(2)

Here, A and B denote the entity sets of the two relations.

First, for highly similar relation pairs, we required Jaccard ≥ 0.8. Second, to handle cases where a low Jaccard score may arise because one triple set is small yet overlaps heavily with another, we additionally required Jaccard ≤ 0.2 and

**Table 3. Detailed list of triple types.**

| No. | Triplet Type | Hetionet | Symmap | TCMbank | STRING | TTD | Total |
|-----|-------------|----------|--------|---------|--------|-----|-------|
| 01 | Anatomy: Gene | 726495 | – | – | – | – | 726495 |
| 02 | Compound: Compound | 6486 | – | – | – | – | 6486 |
| 03 | Compound: DiseaseMM | 1145 | – | – | – | 30312 | 31457 |
| 04 | Compound: Gene | 51429 | – | – | – | – | 51429 |
| 05 | Compound: Side Effect | 138944 | – | – | – | – | 138944 |
| 06 | DiseaseMM: Anatomy | 3602 | – | – | – | – | 3602 |
| 07 | DiseaseMM: DiseaseMM | 543 | – | – | – | – | 543 |
| 08 | DiseaseMM: Gene | 27977 | – | 755977 | – | – | 783954 |
| 09 | DiseaseMM: SymptomMM | 3357 | – | – | – | – | 3357 |
| 10 | Gene: Biological Process | 559504 | – | – | – | – | 559504 |
| 11 | Gene: Cellular Component | 73566 | – | – | – | – | 73566 |
| 12 | Gene: Gene | 474526 | – | – | 1704045 | – | 2178571 |
| 13 | Gene: Molecular Function | 97222 | – | – | – | – | 97222 |
| 14 | Gene: Pathway | 84372 | – | – | – | 8531 | 92903 |
| 15 | Pharmacological Classification: Compound | 1029 | – | – | – | – | 1029 |
| 16 | Herb: Ingredient | – | 88845 | 105749 | – | – | 194594 |
| 17 | Herb: Gene | – | 267804 | 297 | – | – | 268101 |
| 18 | Herb: DiseaseMM | – | – | 181 | – | – | 181 |
| 19 | Ingredient: Gene | – | 29382 | 110109 | – | – | 139491 |
| 20 | Ingredient: DiseaseMM | – | – | 1756 | – | – | 1756 |
| 21 | Gene: Compound | – | – | – | – | 16392 | 16392 |
| 22 | SymptomTCM: SymptomMM | – | 3837 | – | – | – | 3837 |
| 23 | Herb: SymptomTCM | – | 21326 | – | – | – | 21326 |
| 24 | Gene: DiseaseMM | – | 256825 | – | – | 10875 | 267700 |
| 25 | SymptomMM: DiseaseMM | – | 56671 | | – | – | 56671 |
| 26 | Ingredient: SymptomTCM | – | 916780 | – | – | – | 916780 |
| 27 | SymptomMM: Gene | – | 493173 | – | – | – | 493173 |
| 28 | SymptomTCM: Gene | – | 84056 | – | – | – | 84056 |
| 29 | Herb: Syndrome | – | 2110 | – | – | – | 2110 |
| 30 | Ingredient: Syndrome | – | 69115 | – | – | – | 69115 |
| 31 | Syndrome: SymptomTCM | – | 8185 | – | – | – | 8185 |
| 32 | Syndrome: Gene | – | 50193 | – | – | – | 50193 |
| 33 | Total | 2250197 | 2348302 | 974069 | 1704045 | 66110 | 7342723 |

Overlap ≥ 0.6. According to these criteria, we fused the original 58 semantic relations, for example merging "Anatomy: upregulate: Gene" into "Anatomy: express: Gene," and removed five redundant relation types and duplicate triples. Ultimately, the constructed CMKG contains a total of 7,155,373 unique triples.

Due to the massive scale of the CMKG, we conducted a post-hoc stratified manual audit on representative subsets to quantify the integration quality. We prioritized Genes, Diseases, and Herbs for validation, as these entity types represent the most critical semantic bridges between TCM and modern medicine with the highest cross-database overlap. Manual verification of 100 randomly sampled entities across these domains yielded an alignment accuracy of 98.0% (98/100), confirming the robustness of the global name-based ID reassignment. Furthermore, we evaluated the relation fusion logic by sampling 60 triples from the merged and unmerged subsets. The merged subset (e.g., "upregulates" into "expresses") demonstrated a biological rationality rate of 93.3% (28/30), where Anatomy-headed triples showed perfect consistency.

**Table 4. Entity ID naming table.**

| No. | Entity Type | Alignment and Integration | Custom ID |
|---|---|---|---|
| 01 | DiseaseMM | 38346 | DIMM00001-DIMM38271 DM01-DM75 |
| 02 | SymptomMM | 1482 | MS0001-MS1482 |
| 03 | SymptomTCM | 2971 | TS0001-TS2971 |
| 04 | Herb | 7159 | HEB0001-HEB7159 |
| 05 | Ingredient | 46160 | INT00001-INT46160 |
| 06 | Compound | 23637 | DCP00001-DCP23637 |
| 07 | Gene | 26033 | GEE00001-GEE26033 |
| 08 | Pathway | 2208 | PW0001-PW2208 |
| 09 | Molecular Function | 2884 | Retain Source Database ID |
| 10 | Cellular Component | 1391 | Retain Source Database ID |
| 11 | Biological Process | 11381 | Retain Source Database ID |
| 12 | Side Effect | 5701 | Retain Source Database ID |
| 13 | Anatomy | 400 | Retain Source Database ID |
| 14 | Pharmacological Classification | 345 | Retain Source Database ID |
| 15 | Syndrome | 233 | ZH001-ZH233 |

Simultaneously, an audit of semantically similar but unmerged pairs (e.g., "interacts" vs. "experiments") showed an 86.7% (26/30) filtering accuracy in preserving evidence granularity. While minor discrepancies occurred due to inherent database-level scoring biases and the lack of strict ontological constraints for specific predicates, this evaluation confirms the high semantic fidelity of the CMKG.

To comprehensively evaluate the data structure and knowledge coverage of the graph, we further conducted a statistical analysis of the contributions from each data source (see Table 5). Statistical results indicate that the CMKG constructed in this study achieves a multi-level deep integration of "clinical phenotypes—TCM and MM entities—molecular mechanisms". Among them, SymMap (32.51%) and TCMBank (13.03%) form the foundational backbone for cross-system connections, providing abundant mapping relations between TCM herbs, TCM ingredients, and MM targets. On this basis, the introduction of Hetionet (29.72%) and STRING (23.81%) significantly expands the biological depth of the graph, while TTD focuses on providing validated gold-standard support for drug-target and disease-target associations from the perspective of clinical validation data. Specifically, Hetionet supplements macroscopic anatomical and biological process information, whereas STRING provides a high-confidence gene-gene interaction network. This data composition aims to construct a mechanistic reasoning network with a dense topological structure. The introduction of external mechanistic

**Table 5. Data source distribution and contributions in the Cross-Medicine Knowledge Graph (CMKG).**

| Data Source | Original Triples | Fused Triples | Percentage (%) | Key Contribution |
|---|---|---|---|---|
| Symmap | 2348302 | 2326437 | 32.51 | Provides fundamental mapping relationships between TCM herbs, symptoms, and modern medical targets. |
| TCM-bank | 974069 | 932436 | 13.03 | Supplements additional herb entities and traditional formula knowledge. |
| Hetionet | 2250197 | 2126712 | 29.72 | Provides macroscopic associations for anatomy, biological processes, and disease-gene relationships. |
| STRING | 1704045 | 1704045 | 23.81 | Provides high-confidence protein-protein interactions (PPIs). |
| TTD | 66110 | 65743 | 0.92 | Provides gold-standard, validated drug-target and disease-target relationships. |
| Total | 7343723 | 7155373 | 100.00 | Gene-bridged Cross-Medicine Knowledge Graph (CMKG) |

data (Hetionet and STRING), which together account for over 50% of the graph, fills the gap in molecular pathway connectivity prevalent in traditional TCM databases, thereby providing the necessary structural foundation for subsequent graph embedding-based link prediction and path-based interpretable reasoning.

### 3.3 Cross-medicine knowledge graph analysis

Using Cytoscape, we connected entity types across the two medical systems according to their semantic relations to obtain an integrated graph of ancient-and-modern medical entities in Fig 2. The red solid lines indicate that "Gene" has direct semantic links to numerous entities in both TCM and MM, providing a natural bridge for integrating the two systems and tying them more closely through gene targets.

We then quantified entity types with direct semantic links to "Gene" in both systems. A Venn diagram showed 16,045 shared gene targets between TCM and MM. Intersecting these shared targets with the four MM entity types yielded 5,279 targets (32.9% of the shared targets), while intersecting with the four TCM entity types yielded 4,529 targets (28.2% of the shared targets) in Fig 3. These shared gene targets not only connect extensively with diverse entity instances in both systems but also display a relatively balanced distribution. This makes using genes as a bridge to construct a cross-era, cross-system medical knowledge graph more feasible.

## 4. Methodology implementation

### 4.1 Graph embedding models for link prediction

To predict potential therapeutics on our constructed CMKG, we introduced graph embedding techniques. Graph embedding models map entities and relations into a low-dimensional continuous vector space, evaluating the validity probability of candidate triples via a scoring function, $f(h,r,t)$, computed for the head entity ($h$), relation ($r$), and tail entity ($t$), thereby inferring missing links in the graph. Given that the CMKG integrates highly complex semantic associations from

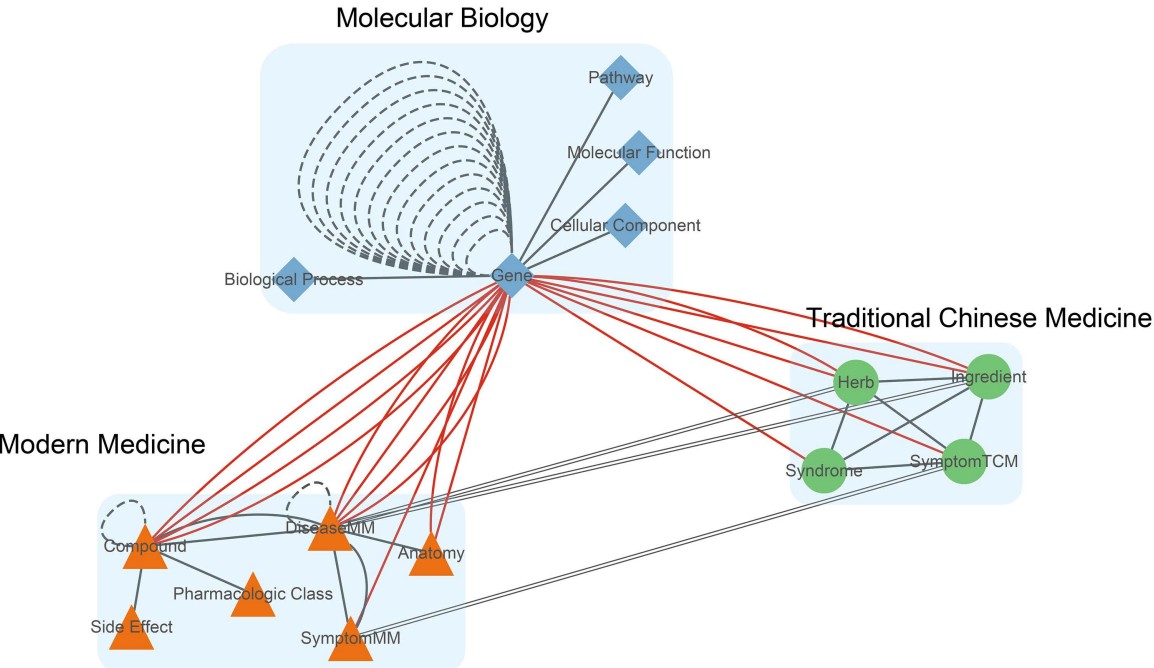

**Fig 2. Association map of TCM and MM entities.**

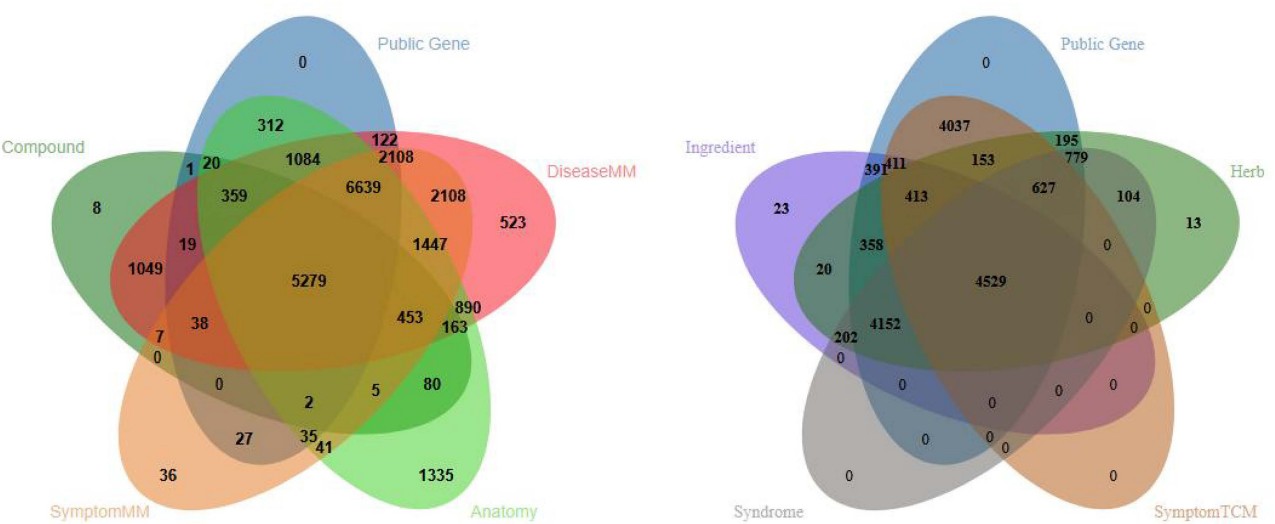

**Fig 3. Intersection of shared gene targets between TCM and MM systems.**

multi-source heterogeneous data across TCM and MM, we trained and evaluated four representative graph embedding models: two classical baselines (TransE, DistMult) and two advanced complex space models (ComplEx, RotatE).

**4.1.1 Classical translational and bilinear baseline models.** As a representative translational distance model, TransE treats relations as translations from head to tail entities in the vector space. Its scoring function is defined as $f(h,r,t) = -\|h + r - t\|$. Although TransE is structurally simple and computationally efficient, the CMKG contains extensive "one-to-many" (e.g., one target corresponding to multiple diseases) and "many-to-many" complex medical mappings, leading to severe representational conflicts in TransE. DistMult is a bilinear product-based model that calculates triple scores by representing relations as diagonal matrices: $f(h,r,t) = \langle h, r, t \rangle = \sum_i h_i r_i t_i$. While the diagonalization significantly reduces parameter complexity, its mathematical properties restrict it to modeling symmetric relations only. Faced with the numerous strictly directional, asymmetric biomedical relations (e.g., "upregulates" or "treats") in the CMKG, DistMult's expressive capacity is markedly limited.

**4.1.2 Complex space embedding models.** To overcome the limitations of the baseline models, we emphasized complex space models that excel in complex networks. ComplEx embeds entities and relations into a complex vector space $C^d$ and ingeniously introduces complex conjugation to break the symmetry constraint of bilinear models. Its scoring function is defined as:

$$f(h,r,t) = Re\left(\langle h, r, \bar{t} \rangle\right)$$

where $h, r, t \in C^d$ are the vector representations of the head entity, relation, and tail entity, respectively; $\bar{t}$ denotes the complex conjugate, and $Re(\cdot)$ represents the real part. Leveraging the flexibility of the complex space, ComplEx performs effectively in modeling symmetric, asymmetric, and transitive relations. In contrast, RotatE models relations as rotations on the complex plane. Specifically, entities are embedded in $C^d$, and relations are constrained to complex vectors with a modulus of 1, applying a rotational transformation to the head entity vector:

$$t = h * r, r \in C^d, |r_i| = 1$$

where ∗ denotes element-wise multiplication. Geometrically, RotatE natively captures symmetric, asymmetric, inverse, and compositional relations, offering a natural advantage in multi-hop reasoning mechanisms.

## 4.2 Experimental settings

For the link prediction task, the fused dataset comprising 7,155,373 triples was randomly partitioned into training (90%), validation (5%), and test (5%) sets. To ensure the rigor of the link prediction task and prevent data leakage, we strictly followed the 'Filtered' evaluation protocol proposed by Bordes et al. [16]. Specifically, during the ranking of a test triple, all other known true triples in the CMKG were removed from the candidate set, ensuring that the model is evaluated on its ability to prioritize truly missing associations rather than recovering existing information. To ensure a fair comparison of all graph embedding models under a unified representational capacity, the core architectural parameters were fixed: the embedding dimension was uniformly set to 400, and the batch size was set to 1024. All models were optimized using a margin-based ranking loss, with 200 negative samples generated for each positive triple. To address potential stochastic variation and balance statistical rigor with maximum predictive sensitivity, we employed a two-stage training and evaluation strategy. In the first stage (Stability Validation), to rigorously evaluate the robustness and relative ranking of the models, all four algorithms were trained for 30 epochs across three independent runs using different random seeds (42, 123, 2026). In the second stage (Full Convergence for Downstream Prediction), we independently optimized the models to full convergence to ensure the highest quality of semantic representations for the subsequent drug discovery task. Due to the diverse spatial mapping mechanisms of the models, they exhibited distinct convergence behaviors. To mitigate loss oscillation and late-stage overfitting—especially for bilinear diagonal models (e.g., DistMult)—independent hyperparameter tuning was conducted. Consequently, the learning rate for ComplEx was set to 1e-3, while RotatE and TransE were fine-tuned to 5e-4. A minimal penalty of $1 \times 10^{-9}$ was employed for TransE, RotatE, and ComplEx to maximize feature retention. For DistMult, a rigorous grid search was performed to find the optimal configuration. Using an early stopping strategy based on validation set saturation, ComplEx achieved convergence at 150 epochs, RotatE and TransE at 100 epochs, whereas DistMult reached its performance peak at epoch 10. This dual-track experimental design effectively ensured both the statistical reliability of model comparison and the empirical optimality for downstream tasks.

## 4.3 Unified path scoring mechanism

While existing link prediction models offer strong performance, their "black-box" nature fails to directly elucidate the specific molecular cascades of drug action. To address this, we combined AnyBURL-based symbolic rule learning with a DFS algorithm to propose a Unified Path Scoring Framework incorporating ingredient specificity and confidence calibration. Unlike traditional path-finding algorithms (such as DeepWalk or topology-based random walks [29]), our framework explicitly introduces a domain-aware semantic weighting mechanism. Traditional methods generally rely solely on path length or node degree, skewing results toward high-frequency, generic nodes and stripping the generated explanatory paths of disease specificity. Our scoring formula innovatively combines an ISI with a hybrid path confidence calibration. This design elevates paths possessing specific biological significance despite having lower frequencies. For any candidate path $P = (e_0, r_1, e_1, r_2, e_2, ..., r_L, e_L)$ from a predicted drug $e_0$ to the target disease, the final score is calculated as:

$$Score_{path} = BaseScore \times Decay(L) \times C_{effective} \times Bonus$$

where the core components are defined as follows:

• Base Semantic Score (BaseScore) and Ingredient Specificity Index (ISI):

The BaseScore is the geometric mean of all relation weights $w_r$ and entity weights $w'_e$ along the path, preventing any single high-weight node from completely dominating. Specifically, a maximum weight of 1.0 is assigned to entity types highly

associated with molecular mechanisms (i.e., Herb, Ingredient, Gene, and Compound) and to inter-gene relations supported by direct or indirect experimental evidence(i.e.,experiments, experiments_transferred). Other inter-gene relations are weighted at 0.9, while the remaining entity and relation types default to a baseline weight of 0.7:

$$BaseScore = \left( \prod_{i=1}^{L} \left( w_{r_i} \cdot w'_{e_i} \right) \right)^{\frac{1}{L}}$$

To counter noise from generic ingredients (e.g., highly ubiquitous entities like quercetin), we drew on the TF-IDF concept from information retrieval to design the ISI. For entities of the "Ingredient" type, the dynamic weight $w'_{e(INT)}$ is calculated as:

$$w'_{e(INT)} = w_{base} \cdot \max \left( 0.01, \frac{\ln \left( N/(df_i+1) \right)}{\ln N} \right)$$

where N is the total number of herb entities in the graph, and $df_i$ is the number of herbs containing ingredient $i$. This effectively suppresses the weight of high-frequency generic components, highlighting nodes with genuine biological specificity.

• Hybrid Path Confidence Calibration

To resolve the confidence space mismatch between deterministic paths discovered by DFS and probabilistic rules mined by AnyBURL, we introduced a calibration factor $\lambda_{DFS}$:

$$C_{effective} = \begin{cases} \min(1.0, C_{rule}), & \text{if the path is derived from a meta-rule ;} \\ \min(1.0, BaseScore \times \lambda_{DFS}), & \text{if the path is discovered by DFS.} \end{cases}$$

Where $C_{rule}$ is the raw confidence of the meta-rule. The hyperparameter $\lambda_{DFS}$ aligns the confidence levels of deterministic search and probabilistic reasoning, preventing artificially high-scoring DFS paths from drowning out high-value logical rules

• Length Decay and Mechanism Bonus:

Following the biomedical heuristic favoring the "shortest path," we capped the maximum path length at 4 hops. The length decay function is defined as: Decay(L) = 1.0 when $L \leq 2$; and Decay(L) = $0.9^{L-2}$ when $2 < L \leq 4$. Furthermore, if a path contains nodes such as Gene or specific Ingredient, or includes relationships with explicit experimental evidence (e.g., "experiments", "experiments_transferred") or the strongly associated TCM component relationship "includes", a global bonus ($Bonus \in [1.2, 1.8]$) is triggered to elevate high-order evidence paths with clear pharmacological mechanisms.

## 5 Results and discussion

### 5.1 Link prediction performance

To validate the effectiveness of the graph embedding models on the CMKG, we trained the four models mentioned above. In link prediction assessments based on knowledge graph embeddings, standard evaluation protocols typically include Mean Reciprocal Rank (MRR) and Hits@K [30]; accordingly, this study adopted these primary metrics.

To rigorously verify the stability of the models against stochastic initialization, the comparative results are reported as Mean ± Standard Deviation based on the multi-seed stability tests (see Table 6). Experimental results indicate that complex space models (ComplEx, RotatE) significantly outperformed traditional baselines with extremely low variance, confirming their robust generalization capabilities. Specifically, TransE encountered severe semantic expressional

**Table 6. Performance comparison of different knowledge graph embedding models on the CMKG dataset.**

| Model | MRR | Hit@1 | Hit@3 | Hit@10 |
|-------|-----|-------|-------|--------|
| TransE | 0.105±0.001 | 0.052±0.001 | 0.119±0.001 | 0.206±0.002 |
| DisMult | < 0.001 | < 0.001 | < 0.001 | < 0.001 |
| RotatE | 0.161±0.001 | 0.097±0.002 | 0.179±0.001 | 0.284±0.001 |
| Complex | **0.213±0.004** | **0.106±0.005** | **0.261±0.005** | **0.418±0.003** |

conflicts when handling the massive "one-to-many" and "many-to-many" medical mappings within the CMKG, yielding a Hits@10 of only 0.206±0.002. DistMult, constrained by the mathematical properties of its symmetric bilinear diagonal matrix, inherently failed to model asymmetric biological relations with clear directionality (e.g., upregulates). This mathematical limitation led to severe loss oscillation and a catastrophic near-zero performance across all metrics, empirically confirming the necessity of introducing complex space representations for our graph. In contrast, ComplEx utilized the complex conjugate mechanism to bypass symmetry constraints, exhibiting exceptional and stable performance in complex multiple-topology matching. Having achieved the optimal comprehensive ranking metrics (MRR=0.213±0.004, Hits@10=0.418±0.003), the statistical superiority of the ComplEx model was firmly established. Based on this, we deployed the fully converged ComplEx model in the second experimental stage. Upon reaching full convergence at 150 epochs, the ComplEx model achieved a peak MRR of 0.26 and a Hits@10 of 0.48. These high-precision representations were then utilized for the subsequent candidate prioritization task to ensure maximum sensitivity and biological accuracy in predicting the therapeutic pathways for T2DM.

## 5.2 Prediction of candidate therapeutics

Based on the best-performing ComplEx model described above, we conducted candidate drug prediction for T2DM (DM06) across the entire graph. To ensure the predictions possessed clear biological significance, we used 709 genes explicitly associated with T2DM in the graph as bridges. We selected six key semantic relations—"Compound: bind: Gene", "Compound: downregulate: Gene", "Compound: upregulate: Gene", "Herb: act_on: Gene", "Gene: is_associated_ with: Compound", and "Herb: is_associated_with: Gene"—as meta-path constraints for the prediction.

The model comprehensively scored and ranked 7,159 TCM herbs and 23,637 modern compounds contained in the graph. According to the link prediction scores, the highest-scoring TCM herb was Abelmoschus manihot (HEB1435), and the highest-scoring modern compound was Topiramate (DCP11264). In the rule learning phase, the AnyBURL tool was utilized. The top 100 TCM herbs and the top 100 modern medicine compounds obtained from the candidate predictions were inserted into the source dataset for rule learning. The maximum path length was restricted to 4 hops, and the learning time was set to 1,000 seconds, with other parameters kept at their default values, to acquire meta-rules mapping drug entities to the disease entity.

## 5.3 Parameter sensitivity analysis

To verify the robustness of the proposed unified path scoring framework and determine the optimal configuration for key hyperparameters, we conducted a two-phase sensitivity experiment. First, we calibrated the DFS penalty factor in the hybrid strategy to balance rule-based and search-based explanatory mechanisms. Second, we evaluated the impact of each scoring function component (weights, decay, bonus, specificity) on the final path rankings.

**5.3.1 Calibration of hybrid path confidence.** We integrated deterministic graph search (DFS) with probabilistic rule reasoning (AnyBURL). However, because DFS paths generally possess higher geometric mean scores, direct mixing often eclipses rule-based paths that have lower confidence but stronger logical foundations. Therefore, we introduced the

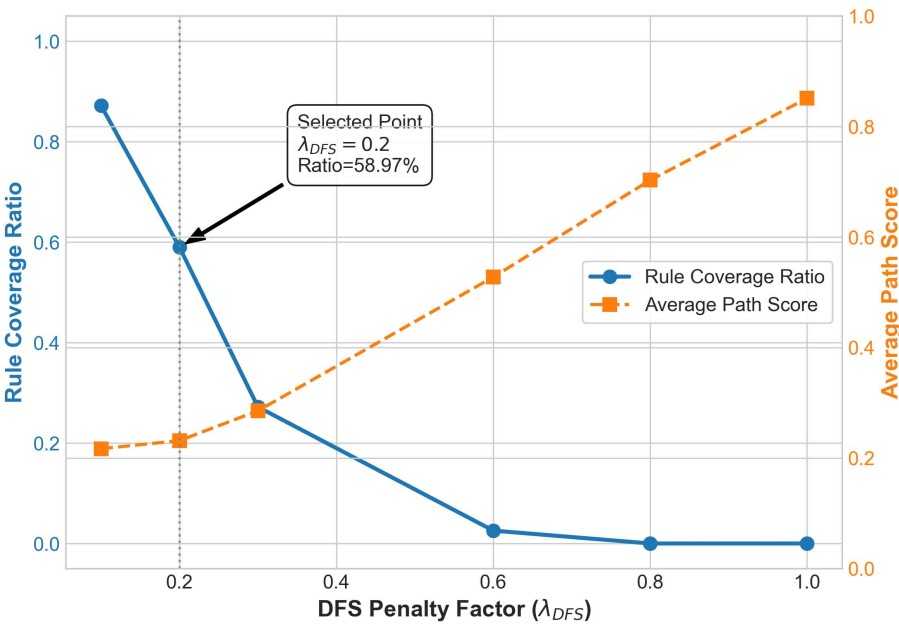

**Fig 4. Impact of the DFS penalty factor $\lambda_{DFS}$ on rule coverage ratio and average path score.**

calibration factor $\lambda_{DFS}$ to penalize pure search paths. Testing $\lambda_{DFS}$ within a range of [0.05, 1.0] regarding its impact on the composition of Top-10 explanatory paths (see Fig 4), we observed:

- When $\lambda_{DFS}$ = 1.0 (no penalty), the Top-10 results were entirely dominated by DFS paths (Rule Ratio = 0%), and the model degraded into a traditional graph traversal.

- When $\lambda_{DFS}$ < 0.1 (heavy penalty), DFS paths were excessively suppressed (Rule Ratio > 80%), stripping the model's ability to discover novel mechanisms outside the rule base.

- The optimal balance was achieved at $\lambda_{DFS}$ = 0.2 (Rule Ratio = 58.97%), retaining logic from AnyBURL while introducing approximately 40% explicit graph paths. Thus, $\lambda_{DFS}$ was fixed at 0.2 for subsequent experiments.

**5.3.2 Global robustness of scoring function.** Building upon the fixed $\lambda_{DFS}$ = 0.2, we further evaluated the stability of the scoring model by altering the configurations of four key dimensions: weight differentiation, length decay, bonus mechanism, and ingredient specificity(see Table 7).

Using Rank Consistency (Intersection over Union, IoU, with the Baseline) as the evaluation metric, we tested the changes in the Top-10 paths for 20 randomly selected drugs. As shown in Fig 5A (left panel), the proposed baseline model maintained its IoU within a reasonable range compared to other variants, demonstrating overall stability without parameter overfitting. In terms of variant impacts, removing all prior entity and relation weights (No Dist.) resulted in the most significant drop in IoU (median < 0.6). This highlights that expert-knowledge-driven weights (e.g., setting Gene to 1.0 versus Anatomy to 0.7) are crucial for filtering high-quality paths. Furthermore, removing the Ingredient Specificity Index (No ISI, i.e., setting it to 1.0) caused the IoU to deviate significantly from 1.0. This deviation confirms that the ISI effectively intervenes in the ranking process, which constitutes a beneficial adjustment.

**Table 7. Hyperparameter settings for sensitivity analysis.**

| Experiment Group | Variant Name | Weight | Decay | Bonus | ISI Lower Bound | Avg IoU (Result) |
|---|---|---|---|---|---|---|
| **Baseline** | **Proposed Model** | **0.7** | **0.9** | **1.8** | **0.01** | **1** |
| Weight | No Distinction | **1.0** | 0.9 | 1.8 | 0.01 | 0.267 |
| | Strong Distinction | **0.2** | 0.9 | 1.8 | 0.01 | 0.861 |
| Length | No Decay | 0.7 | **1.0** | 1.8 | 0.01 | 0.704 |
| | Strong Decay | 0.7 | **0.5** | 1.8 | 0.01 | 0.861 |
| Bonus | No Bonus | 0.7 | 0.9 | **1.0** | 0.01 | 0.792 |
| | High Bonus | 0.7 | 0.9 | **5.0** | 0.01 | 0.850 |
| Specificity | **No ISI (Ablation)** | 0.7 | 0.9 | 1.8 | **1.0** | 0.797 |

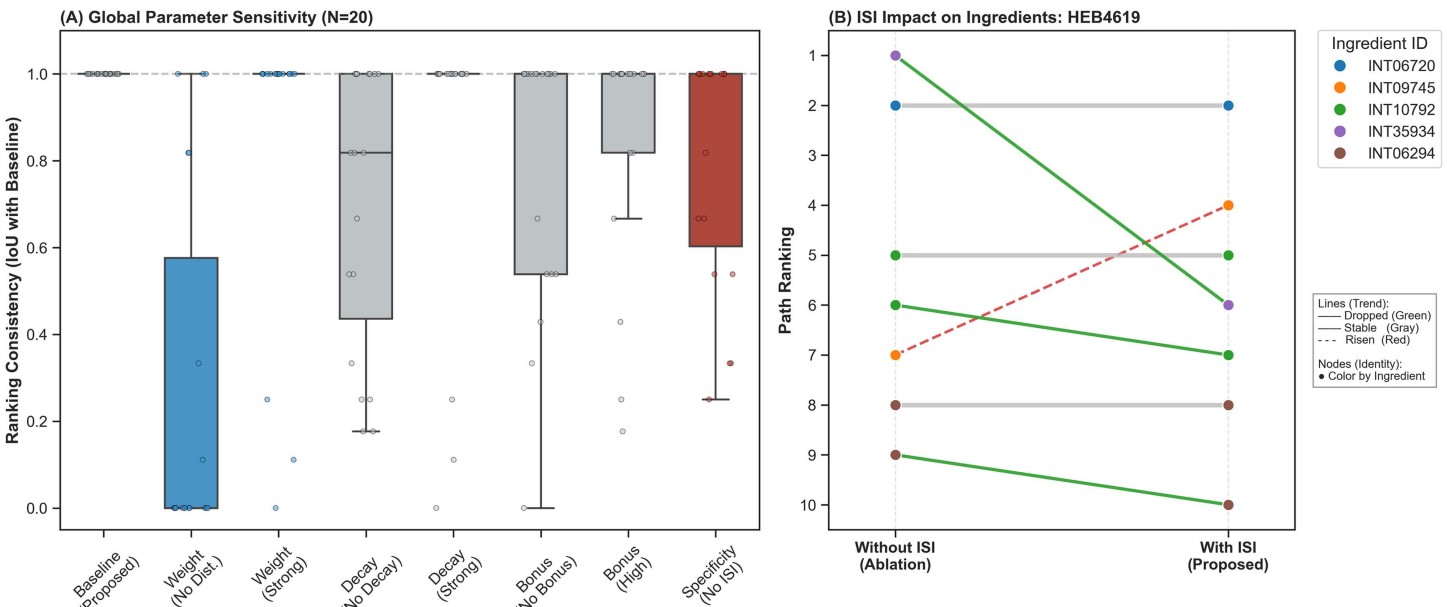

**Fig 5. Global robustness of the scoring function and the specific impact of the Ingredient Specificity Index (ISI).**

To specifically illustrate the role of the ISI, we conducted an in-depth analysis of the ingredient ranking changes before and after its introduction, using Panax Ginseng Leaf (HEB4619)—the herb exhibiting the most substantial ranking shift involving Ingredient entities—as a case study (see Fig 5B, right panel). Without the ISI, generic ingredients such as Palmitic Acid (INT35934) ranked first. However, Palmitic Acid is a ubiquitous fatty acid in flora and fauna, lacking disease specificity. Upon introducing the ISI, its rank dropped to sixth (purple line), successfully down-weighting the entire associated path. Conversely, a more specific ingredient, 58924_FLUKA (INT09745), rose from seventh to fourth place (orange dashed line), indicating that the ISI enables the model to focus on compounds with greater potential pharmacological value. Meanwhile, core ingredients like Dodecane (INT06720) maintained their original rankings, demonstrating that the ISI preserves inherently strong semantic associations. In summary, the parameter sensitivity analysis proves that our framework achieves optimal hybrid explanatory power at $\lambda_{DFS}$ = 0.2, and the ISI mechanism effectively eliminates generic noise while enhancing the biological specificity of the explanatory paths.

## 5.4 Ablation study

The unified path scoring framework, which integrates weights, length decay, the ISI, and a bonus mechanism, is the core module for realizing interpretable candidate drug prioritization in this study. To verify the independent contributions and necessity of each design component within this complex scoring system, we designed four ablation variants (w/o ISI, w/o Weights, w/o Rules, and w/o Bonus) and quantitatively evaluated the Top-10 explanatory paths generated for 20 target drugs. The evaluation employed three core metrics that directly reflect explanation quality: Average Path Specificity (calculated based on the ISI), Evidence Ratio (proportion of edges with experimental evidence), and Rule Contribution. The experimental results (detailed in Table 8 and Fig 6) reveal the critical role of each component within the overall evaluation system.

Specifically, the global weighting and calibration mechanism plays a decisive role. When all prior entity/relation weights and the DFS calibration penalty were removed (w/o Weights), the rule coverage plummeted drastically from 61.58% in the full model to a mere 0.53%. This provides compelling evidence that without the proposed global scoring and calibration mechanism, relying solely on topological search (DFS) would completely drown out the high-value implicit logical rules mined by AnyBURL due to artificially inflated scores. Thus, the weighting system serves as the cornerstone for balancing deterministic search and probabilistic reasoning. Additionally, the bonus mechanism is essential for capturing biological evidence. Removing the reward coefficients for entity types such as "Gene" and relationships including "experiments," "experiments_transferred," and the strongly associated TCM component relationship "includes" (w/o Bonus) caused the Evidence Ratio to plummet from 24.74% to 6.31%, with the Average Path Specificity dropping to its lowest point (0.07). This demonstrates that the bonus mechanism effectively prioritizes hard-core mechanistic paths backed by solid biological experiments (such as protein-protein interactions), thereby greatly enhancing the clinical credibility of the results. Furthermore, the ISI exerts a critical filtering effect. Before introducing the TF-IDF-inspired specificity filter (w/o ISI), the Average

**Table 8. Contribution analysis of model components to knowledge discovery metrics, including average ISI, evidence ratio, and rule alignment.**

| Variant | Avg. ISI | Evid. Ratio | Rule |
|---|---|---|---|
| w/o ISI | 0.24 | 24.02% | 68.42% |
| w/o Weights | **0.31** | 19.11% | 0.53% |
| w/o Rules | 0.25 | 22.50% | 0.00% |
| w/o Bouns | 0.07 | 6.31% | **70.53%** |
| Full Model(Ours) | 0.26 | **24.74%** | 61.58% |

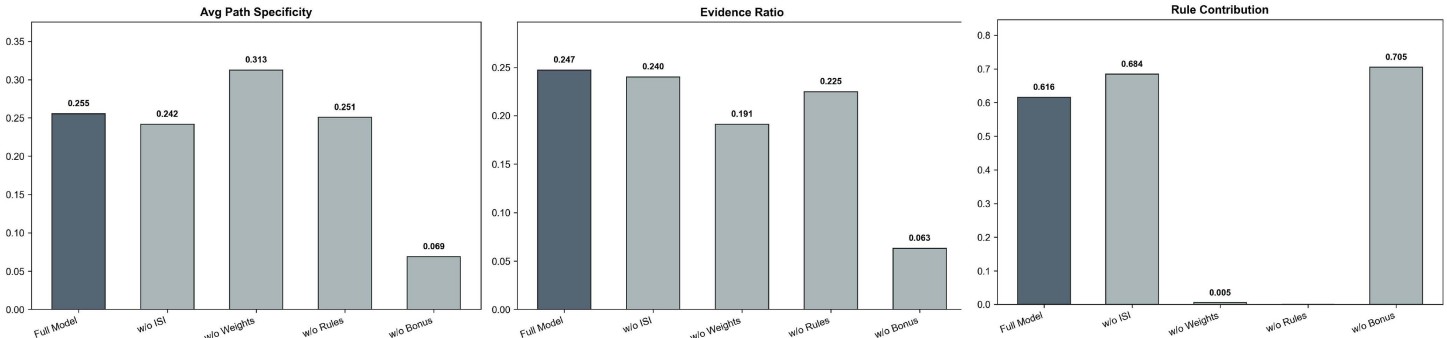

**Fig 6. Quantitative evaluation of ablation variants in the unified path scoring framework.**

Path Specificity score (0.24) was lower than that of the full model (0.26). This corroborates the findings from the earlier case study (Fig 5B), confirming that the ISI mechanism effectively down-weights prevalent, low-value generic ingredients like Palmitic Acid, ensuring that the recommended paths focus on key compounds with higher specificity for T2DM. Finally, the ablation results highlight the complementarity of rules and search. When the rule engine was completely disabled (w/o Rules), the model regressed to a pure graph search, losing the guidance of macroscopic statistical patterns within the graph and failing to establish deep cross-domain logical associations.

In conclusion, the quantitative ablation experiments demonstrate that the proposed unified scoring framework is not a simple aggregation of discrete factors, but a tightly coupled, synergistic organic whole that achieves an optimal global balance across mechanism specificity, biological evidence strength, and logical rule coverage.

### 5.5  Interpretability analysis

Abelmoschus manihot (HEB1435) is utilized in TCM for its efficacy in clearing heat, detoxifying, reducing swelling, and promoting diuresis. Modern pharmacological studies indicate that its rich flavonoid content exhibits significant anti-inflammatory, antioxidant, and renoprotective effects, making it widely used as an adjuvant therapy for diabetic nephropathy [31]. The top 10 key semantic paths for this herb (summarized in Table 9) reveal that its potential mechanisms in intervening with T2DM are primarily concentrated in four interconnected dimensions. First, it regulates insulin signaling (Paths 5 and 6) by modulating the insulin receptor (INSR) and HRAS to intervene with GRB14. As GRB14 is a known negative regulator of insulin signaling, inhibiting its activity has been shown to significantly improve tissue insulin sensitivity and overall glucose homeostasis [32]. Second, the herb exhibits anti-inflammatory effects and aids in the prevention of diabetic complications (Paths 3, 7, and 8). Path 3 involves classical driving factors (PRKCB and IKKβ) that mediate the NF-κB inflammatory pathway, while Paths 7 and 8 highlight the E3 ubiquitin ligase NEDD4 and its interaction network. Recent research indicates that NEDD4 dysfunction contributes to the abnormal degradation of renal ion channels, which is closely linked to the progression of diabetic nephropathy [33]. Third, it protects pancreatic β-cell from stress-induced damage (Path 2) through the regulation of p53 (TP53) and its inducible protein TP53INP1 by MAPK3. Suppressing the overactivated p53-TP53INP1 apoptotic cascade under metabolic stress effectively preserves β-cell mass and maintains insulin secretion [34]. Finally, the paths point to post-transcriptional regulation and RNA splicing (Paths 4 and 10). Nodes such as KHDRBS1 and SON suggest involvement in the RNA alternative splicing network, implying profound epigenetic regulatory potential. Consequently, this framework not only extracts highly specific molecular pathways but also demonstrates that Abelmoschus manihot systematically mitigates T2DM pathology through a multi-target network encompassing insulin sensitization, inflammation control, renal protection, and cell survival.

**Table 9.  Top 10 key semantic paths of Abelmoschus manihot in the treatment of T2DM.**

| No. | Key Semantic Paths | Method | Score |
|---|---|---|---|
| 01 | Abelmoschus manihot→regulate→MAPK3→synergizewith→GAB2→synergizewith→TRAPPC11→act in pathway→T2DM | DFS | 0.2134 |
| 02 | Abelmoschus manihot→inhibit→MAPK3→affect→TP53→induce→TP53INP1→dysregulate→T2DM | DFS | 0.2134 |
| 03 | Abelmoschus manihot→inhibit→PRKCB→synergizewith→IKBKB→synergizewith→SASH1→dysregulate→T2DM | DFS | 0.2134 |
| 04 | Abelmoschus manihot→regulate→RASA1→synergizewith→KHDRBS1→synergizewith→RALY→act in pathway→T2DM | DFS | 0.2134 |
| 05 | Abelmoschus manihot→regulate→RASA1→synergizewith→INSR→bind→GRB14→dysregulate→T2DM | DFS | 0.2134 |
| 06 | Abelmoschus manihot→regulate→RASA1→synergizewith→HRAS→synergizewith→GRB14→dysregulate→T2DM | DFS | 0.2134 |
| 07 | Abelmoschus manihot→regulate→LATS1→synergizewith→ITCH→synergizewith→NDFIP2→dysregulate→T2DM | DFS | 0.2134 |
| 08 | Abelmoschus manihot→regulate→LATS1→synergizewith→NEDD4→synergizewith→NDFIP2→dysregulate→T2DM | DFS | 0.2134 |
| 09 | Abelmoschus manihot→regulate→LATS1→synergizewith→HTRA2→synergizewith→MAEA→act in pathway→T2DM | DFS | 0.2134 |
| 10 | Abelmoschus manihot→regulate→TOP1→synergizewith→ZC3H18→synergizewith→SON→act in pathway→T2DM | DFS | 0.2134 |

Topiramate [35] is an FDA-approved antiepileptic and migraine-prophylaxis drug, with several extended-release and combination formulations approved for clinical use. The top 10 key semantic paths are presented in Table 10.

By integrating evidence from relevant literature and experimental studies, we semantically analyzed the critical reasoning paths through which Topiramate may act as a potential therapeutic agent for T2DM. Paths 1, 2, 3, and 10 involve SMC1A and related factors such as NSMCE2/NSMCE4A, CTCF and BRD4, indicating that topiramate may modulate the cohesin complex and chromatin architecture, indirectly affecting β-cell gene expression and adipose-tissue inflammation, which is consistent with studies on cohesin, CTCF, and BRD4 in metabolic control [36–39]. Paths 4 and 6 point to regulation of CDC20, influencing PPP2R5A-mediated AKT dephosphorylation and downstream p300/CBP function, thereby impacting insulin signaling and diabetic complications [40–43]. Path 5 notes topiramate's inhibition of CYP2C19, potentially shifting metabolic load to CYP2C9 and altering the pharmacokinetics of oral antidiabetic agents [44]. Path 9 suggests binding to CYP3A4 may affect hepatic lipid metabolism and insulin sensitivity via AMFR/gp78-mediated protein degradation [45,46]. At the enzyme and energy-metabolism level, Path 8 shows action on the DLD/DBT complex, perturbing BCAA and pyruvate metabolism and impairing insulin signaling [47–50]. Specifically, Path 7 highlights a neuro-metabolic regulatory axis: evidence suggests that modulating pathways associated with bipolar disorder may indirectly elevate the expression of Parvalbumin (PVALB). As PVALB-positive interneurons are closely linked to systemic energy homeostasis, this path underscores a potential secondary regulatory mechanism through which Topiramate enhances insulin sensitivity [51,52]. While less direct than primary molecular targets, such multi-systemic associations are characteristic of the complex comorbidities between neurological and metabolic disorders. Overall, topiramate's potential benefit may arise from a combination of chromatin regulation, cell-cycle signaling, drug-metabolizing enzyme interactions, and energy-metabolism network effects, jointly shaping T2DM-related phenotypes.

## 5.6 Limitations and future work

Despite the promising results in candidate drug prioritization for T2DM, several limitations of the current study should be acknowledged. First, the proposed framework serves primarily as a hypothesis-generation and prioritization tool rather than a definitive drug discovery platform. Although the identified candidates and mechanistic paths are supported by existing literature, they have not yet undergone rigorous in vitro or in vivo experimental validation. Second, the quality of prioritization depends on the completeness of the underlying biomedical databases. While multi-source integration was performed, inherent noises or missing links in TCM databases may still affect the recall of the model. Future research will focus on incorporating advanced graph representation learning techniques, such as Transformer-based architectures, and integrating experimental validation to further verify the anti-diabetic effects of the prioritized candidates.

**Table 10. Top 10 key semantic paths of Topiramate in the treatment of T2DM.**

| No. | Key Semantic Paths | Method | Score |
|---|---|---|---|
| 01 | Topiramate→inhibit→SMC1A→synergize with→NSMCE4A→act in pathway→T2DM | Rule | 0.3924 |
| 02 | Topiramate→inhibit→SMC1A→synergize with→NSMCE2→act in pathway→T2DM | Rule | 0.3189 |
| 03 | Topiramate→inhibit→SMC1A→synergize with→CTCF→act in pathway→T2DM | Rule | 0.3118 |
| 04 | Topiramate→inhibit→CDC20→synergize with→PPP2R5A→dysregulate→T2DM | Rule | 0.2736 |
| 05 | Topiramate→bind→CYP2C19→synergize with→CYP2C9→act in pathway→T2DM | Rule | 0.2718 |
| 06 | Topiramate→inhibit→CDC20→synergize with→EP300→dysregulate→T2DM | Rule | 0.2428 |
| 07 | Topiramate→alleviate→Bipolar Disorder→elevate→PVALB→dysregulate→T2DM | Rule | 0.2402 |
| 08 | Topiramate→inhibit→DLD→synergize with→DBT→act in pathway→T2DM | Rule | 0.2391 |
| 09 | Topiramate→bind→CYP3A4→degrade→AMFR→act in pathway→T2DM | Rule | 0.2316 |
| 10 | Topiramate→inhibit→SMC1A→synergize with→BRD4→act in pathway→T2DM | Rule | 0.2305 |

## 6. Conclusion

Using genes as a bridge, this study achieved deep semantic interconnection and fusion across TCM and MM systems. By comprehensively collecting and systematically analyzing entity types directly associated with genes in both TCM and MM, we identified 16,045 shared gene targets. The intersections of these shared targets with MM and TCM entity types accounted for 32.9% and 28.2%, respectively. This result not only validates the feasibility of using gene targets to mediate semantic alignment between TCM and MM but also bridges the gap in molecular mechanism connectivity within traditional TCM databases, laying a solid data foundation for constructing a high-resolution mechanistic reasoning network.

Building on this, we proposed an interpretable drug prediction framework coupling graph embeddings with rule learning. Comparative experiments demonstrated that when faced with the massive "one-to-many" and asymmetric biological associations in the CMKG, classical translational models (TransE) and bilinear models (DistMult) easily fall into semantic conflicts or loss oscillation. Conversely, complex space models (ComplEx and RotatE) better capture complex topologies, with ComplEx achieving optimal performance on comprehensive ranking metrics (MRR and Hits@10) due to its complex conjugate properties. However, pure graph embedding methods still suffer from "black-box" limitations. To overcome this, our study integrated AnyBURL rule mining with DFS, innovatively proposing a unified path scoring mechanism. This mechanism not only synthesizes topological decay and a priori weights but also introduces an Ingredient Specificity Index (ISI) to effectively filter high-frequency generic noise. Furthermore, it employs a confidence calibration factor, $\lambda_{DFS}$, to perfectly balance the weight differences between deterministic search and probabilistic reasoning. Empirical research using T2DM as a case study showed that this mechanism successfully translates probabilistic model predictions into highly biologically credible molecular cascades. Rigorous ablation studies and parameter sensitivity analyses further confirmed that this scoring framework maintains exceptional robustness across parameter variations, significantly enhancing the specificity and transparency of knowledge discovery.

Despite demonstrating promising application prospects, this framework still has certain limitations. First, model performance is highly dependent on the quality and coverage of the underlying databases (e.g., SymMap, STRING); inherent missing data may limit the recall of predictions. Second, although the hyperparameters in the scoring formula underwent sensitivity optimization, they still embed certain a priori expert assumptions. Reducing reliance on manual parameter tuning is a critical optimization direction for the future. Future work will focus on further improving the model's representation learning capabilities and automated mechanisms. On one hand, we plan to explore adaptive weight learning and multi-objective optimization algorithms to dynamically replace fixed hyperparameters. On the other hand, given the excellent interpretability and generalization capabilities demonstrated by the Transformer-based graph representation learning model proposed by Su et al. [53] in pan-cancer networks, as well as the latest progress by Yang et al. [54] on link-based attributed graph clustering algorithms, we intend to introduce Transformer architectures and attributed graph generative clustering techniques into the cross-medicine graph. This will facilitate a deeper excavation of high-order topological features among TCM and MM entities, leveraging the semantic enhancement of Large Language Models (LLMs) to achieve dynamic graph completion. Ultimately, this will help build a more intelligent and adaptive ecosystem for integrative medicine drug discovery.

## Acknowledgments

We thank Professor Yonghong Xie for providing conceptual inspiration on the overall study design during the initial drafting of this work.We also thank Dr. Qin Li for her explanations and assistance regarding domain knowledge related to Traditional Chinese Medicine entities.

## Author contributions

**Conceptualization:** Zekun Zhou, Shuo Yang.

**Data curation:** Zekun Zhou.

**Formal analysis:** Zekun Zhou.

**Funding acquisition:** Shuo Yang.

**Methodology:** Zekun Zhou, Shuo Yang.

**Project administration:** Shuo Yang.

**Supervision:** Shuo Yang.

**Visualization:** Zekun Zhou.

**Writing – original draft:** Zekun Zhou.

**Writing – review & editing:** Zekun Zhou, Shuo Yang.

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
