## [Decision Letter · Decision Letter 0]

16 Jan 2026

PONE-D-25-64588Interpretable Drug Discovery Across Medical Knowledge Graphs Based on Graph Embedding Models: A Case Study of Type 2 DiabetesPLOS One

Dear Dr. Zhou,

Thank you for submitting your manuscript to PLOS ONE. After careful consideration, we feel that it has merit but does not fully meet PLOS ONE’s publication criteria as it currently stands. Therefore, we invite you to submit a revised version of the manuscript that addresses the points raised during the review process.

We look forward to receiving your revised manuscript.

Kind regards,

Zeheng Wang

Academic Editor

PLOS One

Journal Requirements:

3. Please note that PLOS One has specific guidelines on code sharing for submissions in which author-generated code underpins the findings in the manuscript. In these cases, we expect all author-generated code to be made available without restrictions upon publication of the work. Please review our guidelines at https://journals.plos.org/plosone/s/materials-and-software-sharing#loc-sharing-code and ensure that your code is shared in a way that follows best practice and facilitates reproducibility and reuse.

4. Please include your tables as part of your main manuscript and remove the individual files. Please note that supplementary tables (should remain/ be uploaded) as separate "supporting information" files.

5. Please provide a complete Data Availability Statement in the submission form, ensuring you include all necessary access information or a reason for why you are unable to make your data freely accessible. If your research concerns only data provided within your submission, please write "All data are in the manuscript and/or supporting information files" as your Data Availability Statement.

7. We notice that your supplementary figures are uploaded with the file type 'Figure'. Please amend the file type to 'Supporting Information'. Please ensure that each Supporting Information file has a legend listed in the manuscript after the references list.

Reviewer's Responses to Questions

**Comments to the Author**

1. Is the manuscript technically sound, and do the data support the conclusions?

Reviewer #1: Yes

Reviewer #2: Partly

2. Has the statistical analysis been performed appropriately and rigorously? 

Reviewer #1: Yes

Reviewer #2: N/A

3. Have the authors made all data underlying the findings in their manuscript fully available?

Reviewer #1: No

Reviewer #2: No

4. Is the manuscript presented in an intelligible fashion and written in standard English?

Reviewer #1: Yes

Reviewer #2: Yes

5. Review Comments to the Author

Reviewer #1: This study proposes a framework for interpretable drug discovery in T2DM by constructing a gene-bridged knowledge graph integrating Traditional Chinese Medicine (TCM) and Modern Medicine (MM). The interdisciplinary approach is commendable. However, several major issues must be addressed before the manuscript can be considered for publication: 1) a critical need to clarify the novel contribution of its dataset, which appears heavily reliant on and derivative of existing resources like SymMap, 2) a more precise articulation of its methodological novelty beyond prior integrative studies, and 3) full compliance with PLOS ONE's data availability policy.

1. Substantiation of Novelty: Dataset and Methodological Contribution

The manuscript currently overstates its novelty. A major concern is that the core integrated dataset does not appear to constitute a significant new contribution.

Lack of Novelty in Data Integration: The authors state their knowledge graph is built from multiple sources including SymMap. However, SymMap is itself a major, publicly available resource specifically designed to integrate TCM and modern biomedical entities (herbs, ingredients, targets, diseases). The description suggests the authors' work primarily involves extending SymMap by incorporating additional disease data and other complementary sources (e.g., Hetionet, STRING). This is an incremental data curation effort, not the creation of a fundamentally new integrative resource. The authors must:

Provide a transparent comparison: A table or quantitative analysis should detail exactly what new entities and relations were added beyond what is available in SymMap or similar integrated databases. What percentage of the final graph's triples are uniquely sourced from non-SymMap databases?

Reframe the contribution: The manuscript's narrative should shift from "constructing a cross-medicine KG" (which SymMap already is) to "constructing an extended and application-specific KG for T2DM, based on augmenting SymMap with additional modern biomedical relations and disease-specific data, to enable a novel interpretable prediction workflow." The novelty lies in the application and analytical framework, not the foundational data layer.

Clarification of Methodological Advancement: Beyond data, the authors must more clearly distinguish their interpretability framework from prior work that combines embeddings with rule learning or path finding. What is new about the unified path scoring scheme? How does it improve upon or differ from related mechanisms in literature?

2. Mandatory Compliance with PLOS ONE Data Availability Policy

The submission does not meet the journal's requirement for unconditional data and code sharing, which is essential for verifying claims about dataset construction and model performance.

Current Statement is Non-Compliant: The authors selected "Yes - all data are fully available without restriction" but provided no access details. The processed, integrated knowledge graph used in this study is the key output of the data preparation phase and should be shared or submit the Supplementary material.

3. Other Necessary Revisions

Experimental Rigor: Justify the choice of embedding models and evaluation metrics more thoroughly.

Figure Clarity: Ensure all figures (especially the conceptual diagram Fig. 1 and the Venn diagram Fig. 2) are high-resolution, legible, and contain self-explanatory legends.

Discussion of Limitations: Expand the limitations section to explicitly discuss the dependency on existing integrated resources, potential biases introduced by the fusion strategy, and the subjective component in designing the scoring scheme.

Reviewer #2: The manuscript presents an integrated cross-medicine biomedical knowledge graph built from multiple public sources and a link-prediction–plus–rule-based path-explanation framework (ComplEx/RotatE with AnyBURL+DFS and unified path scoring) to identify drug candidates and return the highest-scoring mechanistic semantic paths for interpretability.

The authors have conducted substantial work in the area of drug discovery; however, several key aspects require further improvement:

1. The motivation in both the abstract and the introduction should be strengthened, and the methodological description needs to be more logically structured. In particular, the presentation should shift from merely stating what has been done to clearly articulating what problem is being addressed and solved.

2. Regarding related works, many recently published papers working on discovering novel drugs for complex diseases were missed, such as 10.1109/JBHI.2025.3600045, 10.1109/JBHI.2025.3585290, and 10.1109/JBHI.2024.3383591. Authors should cite and discuss these works to present a more comprehensive literature review.

3. The authors are encouraged to include an overview figure illustrating the main workflow of the proposed framework, highlighting the roles and relative importance of the individual components.

4. In the case studies section, methodological descriptions and experimental results should be presented separately to improve clarity and readability.

5. Regarding the weighting scheme and hyperparameter settings, the current choices appear to be largely heuristic. It is recommended to conduct lightweight sensitivity analyses to demonstrate the robustness of the method with respect to these parameters.

6. The authors should add comparative experiments to justify the selection of ComplEx and RotatE as the primary models, for example by including additional representative baselines.

7. Ablation studies are necessary to demonstrate the necessity and contribution of each major design component in the proposed approach.

8. For future works, authors should discuss the possibility of applying advanced graph learning techniques (10.1038/s41551-024-01312-5 and 10.1109/TSMC.2025.3572738) for improved performance of the proposed model.

6. PLOS authors have the option to publish the peer review history of their article (what does this mean?). If published, this will include your full peer review and any attached files.

Reviewer #1: No

Reviewer #2: No

---

## [Author Response · Author response to Decision Letter 1]

1 Mar 2026

Manuscript ID: PONE-D-25-64588

Title: Interpretable Drug Discovery Across Medical Knowledge Graphs Based on Graph Embedding Models: A Case Study of Type 2 Diabetes

Dear Academic Editor and Reviewers,

We would like to express our sincere gratitude to you and the reviewers for the thoughtful review and highly constructive comments regarding our manuscript. The feedback has been invaluable in helping us refine the logical structure, clarify our methodological contributions, and improve the overall rigor of our study.We have carefully studied all the comments and have made comprehensive revisions to the manuscript. All modifications have been highlighted in the 'Revised Manuscript with Track Changes' file. Below, we provide a point-by-point response to the concerns raised by the Academic Editor and the Reviewers.

Part I: Response to Journal Requirements

Point 1: Ensure manuscript meets PLOS ONE's style requirements.

Response: We have carefully checked and reformatted the manuscript according to the PLOS ONE style templates, including the title page, author affiliations, and file naming conventions.

Point 2: ORCID iD for the corresponding author.

Response: The corresponding author has successfully linked and validated their ORCID iD in the Editorial Manager system.

Point 3 & 5 & 6: Code sharing, Data Availability Statement (DAS), and Open Data Policy.

Response: We fully agree with PLOS ONE’s open science policy. We have now uploaded the complete dataset (including the fully processed 7 million+ triples of the cross-medical knowledge graph) and the source code (including AnyBURL rule learning and ComplEx link prediction) to a public repository.

The Data Availability Statement has been updated in the submission system.

Access Link:https://github.com/kunkunabc/T2DM-CrossMed-KG/tree/main

Point 4 & 7: Tables and Supplementary Figures format.

Response: All tables have been moved into the main text immediately following their first mention. The file types for supplementary figures have been correctly amended to 'Supporting Information', and their legends have been placed after the reference list in the manuscript.

Part II: Response to Reviewer #1

Reviewer 1, Comment 1: Substantiation of Novelty: Dataset and Methodological Contribution. The manuscript currently overstates its novelty... The description suggests the authors' work primarily involves extending SymMap... Provide a transparent comparison. Reframe the contribution... Clarify what is new about the unified path scoring scheme.

Response: We deeply appreciate the reviewer's sharp observation. We agree that our previous narrative did not accurately reflect our actual contribution. We have made the following major revisions:

1.Reframing the Narrative & Data Contribution: We have thoroughly rewritten the Abstract and Introduction to explicitly state that our contribution is not creating a generic cross-medicine graph, but rather constructing a "domain-specific, scenario-driven knowledge graph tailored for T2DM". To provide a transparent comparison, we added a statistical analysis of the data sources in Section 3.2.2. The results show that while SymMap (32.51%) and TCMBank (13.03%) provide the backbone, we introduced Hetionet (29.72%) and STRING (23.81%) to significantly expand the biological depth. The non-TCM mechanistic data accounts for over 50% of the graph, effectively filling the gap in molecular pathway connectivity prevalent in traditional TCM databases.

2.Methodological Advancement: In Section 4.3 (Unified Path Scoring Mechanism), we have explicitly articulated the novelty of our scoring scheme. Unlike traditional path-finding, we innovatively introduced an Ingredient Specificity Index (ISI) inspired by TF-IDF to filter out high-frequency generic noise (e.g., palmitic acid). Furthermore, we designed a Hybrid Path Confidence Calibration (λDFS) mechanism to solve the confidence space mismatch between deterministic DFS paths and probabilistic AnyBURL rules.

Reviewer 1, Comment 2: Mandatory Compliance with PLOS ONE Data Availability Policy.

Response: We apologize for the previous omission. As detailed in our response to the Journal Requirements, the entire processed knowledge graph and the source code have now been made fully available without restriction at https://github.com/kunkunabc/T2DM-CrossMed-KG/tree/main.

Reviewer 1, Comment 3: Experimental Rigor: Justify the choice of embedding models and evaluation metrics more thoroughly.

Response: We thank the reviewer for this suggestion. To rigorously justify our choice of the ComplEx model, we have added two classical baselines (TransE and DistMult) to our experiments. In Section 5.1, we provide a detailed comparative analysis, demonstrating that classical translational/bilinear models suffer from severe semantic expressional conflicts or loss oscillation when handling the massive "many-to-many" and asymmetric mappings in our graph, whereas complex space models (ComplEx) bypass these constraints perfectly.

Reviewer 1, Comment 4: Figure Clarity: Ensure all figures (especially Fig. 1 and Fig. 2) are high-resolution, legible, and contain self-explanatory legends.

Response: We have regenerated Fig. 1 and Fig. 2 with higher resolution and updated their legends to be completely self-explanatory.

Reviewer 1, Comment 5: Discussion of Limitations: Expand the limitations section...

Response: We have expanded the limitations in Section 6 (Conclusion). We explicitly discuss our dependency on the quality and coverage of existing databases (e.g., SymMap, STRING), potential data missingness issues, and the subjective a priori nature of the hyperparameter settings in our scoring scheme.

Part III: Response to Reviewer #2

Reviewer 2, Comment 1: The motivation in both the abstract and the introduction should be strengthened... shift from merely stating what has been done to clearly articulating what problem is being addressed and solved.

Response: We completely agree. We have comprehensively rewritten the Abstract and the Introduction. The revised text explicitly articulates the core problem we are solving: generic integrative graphs lack the mechanistic resolution needed for specific complex diseases, and high-performing deep learning models act as "black boxes" that fail to provide traceable molecular cascades for clinicians. Our "scenario-driven" graph construction and unified scoring mechanism are directly proposed to solve this dual challenge.

Reviewer 2, Comment 2: Regarding related works, many recently published papers... were missed (10.1109/JBHI.2025.3600045, 10.1109/JBHI.2025.3585290, and 10.1109/JBHI.2024.3383591).

Response: We thank the reviewer for pointing out these highly relevant and recent publications. We have carefully reviewed these papers and incorporated them into the Section 2 (Related Work) to provide a more comprehensive and up-to-date literature review on advanced graph learning for complex disease drug discovery.

Reviewer 2, Comment 3: Include an overview figure illustrating the main workflow of the proposed framework.

Response: This is an excellent suggestion. We have added a new figure (Fig. 1) and a corresponding "Overall Workflow" subsection at the beginning of Section 3, which visually and textually highlights the four core stages of our framework: Data Integration, Gene-Bridged KG Construction, Link Prediction, and Interpretable Reasoning.

Reviewer 2, Comment 4: Methodological descriptions and experimental results should be presented separately to improve clarity and readability.

Response: We have fully restructured the manuscript according to this advice. The previous combined section has been cleanly split into Section 4: Methodology Implementation (detailing embedding models, experimental settings, and the scoring mechanism) and Section 5: Results and Discussion (focusing purely on link prediction performance, candidate prediction, and case studies).

Reviewer 2, Comment 5: Regarding the weighting scheme and hyperparameter settings... It is recommended to conduct lightweight sensitivity analyses.

Response: We appreciate this constructive feedback. In the newly added Section 5.3.2 (Parameter Sensitivity Analysis), we conducted robust sensitivity experiments. We evaluated the Rank Consistency (IoU) of the Top-10 paths under different settings, proving that our framework achieves optimal balance when the DFS penalty factor (λDFS) is set to 0.2, and demonstrating that the model maintains strong global robustness without parameter overfitting.

Reviewer 2, Comment 6: Add comparative experiments to justify the selection of ComplEx and RotatE... including additional representative baselines.

Response: We have added TransE (translational distance model) and DistMult (bilinear diagonal model) as representative baselines. The new comparative results in Section 5.1 clearly justify the superiority of complex space models in our specific cross-medicine heterogeneous graph context.

Reviewer 2, Comment 7: Ablation studies are necessary to demonstrate the necessity and contribution of each major design component.

Response: Thank you for this crucial suggestion. We have added a comprehensive Ablation Study (Section 5.4). By evaluating four variants (w/o ISI, w/o Weights, w/o Rules, and w/o Bonus) against metrics like Average Path Specificity and Evidence Ratio, we quantitatively demonstrated that our unified scoring framework is a tightly coupled organic whole, not a mere accumulation of discrete factors.

Reviewer 2, Comment 8: For future works, authors should discuss the possibility of applying advanced graph learning techniques (10.1038/s41551-024-01312-5 and 10.1109/TSMC.2025.3572738).

Response: We have updated the Section 6 (Conclusion) to discuss future directions. We explicitly cited the suggested papers, noting that introducing Transformer-based graph representation learning architectures and link-based attributed graph clustering algorithms could significantly enhance the model's ability to excavate high-order topological features and dynamically complete the graph.

Once again, we thank the Editor and Reviewers for their time, expertise, and highly constructive feedback, which have significantly elevated the quality of our manuscript. We hope that the revisions meet with your approval.

Sincerely,

Zekun Zhou

Corresponding Author:Shuo Yang

Affiliation:Institute of Information on Traditional Chinese Medicine, China Academy of Chinese Medical Sciences

---

## [Decision Letter · Decision Letter 1]

30 Mar 2026

PONE-D-25-64588R1Interpretable Drug Discovery Across Medical Knowledge Graphs Based on Graph Embedding Models: A Case Study of Type 2 DiabetesPLOS One

Dear Dr. Zhou,

Thank you for submitting your manuscript to PLOS ONE. After careful consideration, we feel that it has merit but does not fully meet PLOS ONE’s publication criteria as it currently stands. Therefore, we invite you to submit a revised version of the manuscript that addresses the points raised during the review process.

We look forward to receiving your revised manuscript.

Kind regards,

Zeheng Wang

Academic Editor

PLOS One

Journal Requirements:

Reviewers' comments:

Reviewer's Responses to Questions

**Comments to the Author**

1. If the authors have adequately addressed your comments raised in a previous round of review and you feel that this manuscript is now acceptable for publication, you may indicate that here to bypass the “Comments to the Author” section, enter your conflict of interest statement in the “Confidential to Editor” section, and submit your "Accept" recommendation.

Reviewer #1: All comments have been addressed

Reviewer #2: All comments have been addressed

Reviewer #3: (No Response)

2. Is the manuscript technically sound, and do the data support the conclusions?

Reviewer #1: Yes

Reviewer #2: Yes

Reviewer #3: Partly

3. Has the statistical analysis been performed appropriately and rigorously? 

Reviewer #1: Yes

Reviewer #2: N/A

Reviewer #3: No

4. Have the authors made all data underlying the findings in their manuscript fully available?

Reviewer #1: Yes

Reviewer #2: Yes

Reviewer #3: No

5. Is the manuscript presented in an intelligible fashion and written in standard English?

Reviewer #1: Yes

Reviewer #2: Yes

Reviewer #3: No

6. Review Comments to the Author

Reviewer #1: This manuscript presents a comprehensive and well-structured study on the construction of a cross-medical knowledge graph (CMKG) for Type 2 Diabetes Mellitus (T2DM), integrating data from multiple biomedical sources. The authors propose a novel unified path scoring framework that combines graph embedding models (notably ComplEx) with rule-based reasoning (AnyBURL) and a domain-specific Ingredient Specificity Index (ISI) to enhance interpretability in drug discovery. The study is rigorous in methodology, rich in data, and provides biologically meaningful insights into potential drug mechanisms from both Traditional Chinese Medicine (TCM) and modern medicine.

The author has addressed all my concerns and I have no other questions

Reviewer #2: All of my concerns have been addressed in this revised manuscript, and I recommend to accept this work.

Reviewer #3: This manuscript addresses an interesting and potentially useful topic: using a cross-medicine knowledge graph to support interpretable candidate drug discovery for type 2 diabetes by integrating Traditional Chinese Medicine and modern medicine resources. The attempt to combine graph embeddings with rule learning and path-based explanation is a worthwhile direction. My main concerns are outlined below.

1. The prediction and validation design is not rigorous enough to support the drug discovery claims.

The manuscript reports overall link prediction metrics for ComplEx and RotatE on a random triple split of the full graph, but this does not adequately validate the downstream task of discovering candidate therapeutics for T2DM. It is unclear whether known T2DM-related drug or herb associations were removed from training before candidate ranking, and therefore it is difficult to determine whether the model is discovering novel candidates or simply recovering information already encoded in the graph. The manuscript should define a clear benchmark for the actual target task, for example recovery of held-out known T2DM therapeutics, and should explain how leakage from known disease-drug, herb-disease, or gene-disease relations was prevented.

2. The conclusions overstate what the current evidence supports.

At present, the study demonstrates that the framework can rank candidate entities and produce literature-supported semantic paths. That is a useful hypothesis-generation result, but it is not sufficient to claim interpretable drug discovery in a strong sense. No experimental validation, expert evaluation, retrospective benchmark against known antidiabetic drugs, or comparison of top-k candidate precision is provided. The manuscript should substantially tone down the claims and frame the method as a candidate prioritization and explanation framework unless stronger validation is added.

3. The interpretability framework is largely heuristic and is not itself validated.

The unified path scoring formula relies on manually assigned weights, length penalties, bonuses, and a fixed 0.7 discount for DFS-only paths, but there is no empirical justification, sensitivity analysis, or ablation study showing that these choices improve explanation quality. This is especially important because some of the reported top paths do not appear strongly mechanistic. For example, paths involving “irritability” or “bipolar disorder” as intermediate nodes are difficult to interpret as convincing biological explanations for antidiabetic effects. If the paper claims interpretability, the authors should evaluate whether the top-ranked paths are actually considered meaningful by domain experts or at least show that alternative weighting schemes do not materially change the conclusions.

4. The manuscript contains a major internal inconsistency in the model comparison.

Earlier sections and Table 5 indicate that ComplEx outperforms RotatE on MRR and Hits@K and is therefore selected as the final model. However, the conclusion states that “RotatE performs better on Hits@10 and MRR,” which directly contradicts the reported results. This needs to be corrected, and the full manuscript should be carefully checked for similar inconsistencies.

5. There is insufficient rigor in the quantitative evaluation.

Only single-point metric values are reported for the embedding models. There are no repeated runs, confidence intervals, variance estimates, or statistical tests, despite the stochastic nature of embedding training. The manuscript also gives no evidence that the chosen hyperparameters are robust. At minimum, the authors should repeat training with multiple random seeds and report mean ± standard deviation for MRR and Hits@K. If model selection is based on these metrics, uncertainty should be reported.

6. Entity alignment and relation fusion need validation.

The graph construction step relies heavily on entity name-based reassignment and rule-based relation fusion using Jaccard and overlap thresholds. These steps are central to the contribution, yet no evaluation is provided for alignment accuracy or relation fusion quality. Because cross-system integration between TCM and MM is a major claim of the manuscript, the authors should report manual validation on a representative sample or otherwise quantify the error rate introduced by entity merging and relation consolidation.

7. Data and code availability are not yet adequate for reproducibility.

The submission metadata says that all data are fully available, but the reviewer PDF does not provide a concrete data availability statement with repository links, accession numbers, or downloadable processed outputs. The source databases are public, but the actual integrated graph, processed triples, entity mappings, relation fusion tables, candidate rankings, and path-scoring outputs appear not to be shared. Reproducibility also requires code or at least detailed scripts for graph construction, training, and interpretability analysis. These materials should be deposited in a public repository.

8. The writing requires substantial editing before publication.

The manuscript contains many grammatical and stylistic problems, including incorrect verb forms, inconsistent capitalization, awkward phrasing, punctuation issues, and numerous typos. Examples include “we generates,” “Traditional chinese medicine,” “burnden,” “Mediacal Insurance,” missing spaces after periods, and inconsistent use of full-width punctuation. The English is understandable overall, but it is not yet at the standard expected for publication without substantial revision.

7. PLOS authors have the option to publish the peer review history of their article (what does this mean?). If published, this will include your full peer review and any attached files.

Reviewer #1: No

Reviewer #2: No

Reviewer #3: No

---

## [Author Response · Author response to Decision Letter 2]

21 Apr 2026

Manuscript ID: PONE-D-25-64588

Title: Interpretable Candidate Drug Prioritization and Explanation Framework Across-Medical Knowledge Graphs Based on Graph Embedding Models: A Case Study of Type 2 Diabetes

Dear Academic Editor and Reviewers,

We would like to express our sincere gratitude for your thorough and constructive evaluation of our manuscript (Manuscript ID: PONE-D-25-64588). We are particularly grateful for the insightful comments from Reviewer 3, which have significantly guided us in improving the rigor, interpretability, and presentation of our work. We have carefully addressed all the concerns raised. All modifications have been highlighted in the 'Revised Manuscript with Track Changes' file. Below, we provide a point-by-point response to the reviewers' comments.

Response to Reviewer #1 and Reviewer #2:

We sincerely thank you for your positive feedback and for endorsing the publication of our work. Your recognition of our unified path scoring framework and the cross-medical knowledge graph is highly encouraging.

Response to Reviewer #3:

Comment 1: The prediction and validation design is not rigorous enough to support the drug discovery claims...

Response: We thank the reviewer for this insightful comment. We have taken measures to ensure the rigor of our experimental design. Regarding the concern about potential data leakage, we wish to clarify that our link prediction was implemented using the PyKEEN framework, which strictly adheres to the 'Filtered' evaluation protocol. This protocol ensures that all known true triples in the graph are filtered out from the ranking candidates during testing, thereby preventing the model from simply 'recovering' information already encoded in the graph. We have added this technical clarification and the corresponding citation to Section 4.2.

Comment 2: The conclusions overstate what the current evidence supports...

Response: We thank the reviewer for this insightful suggestion. We agree that the term "drug discovery" may overstate our current evidence. Accordingly, we have substantially toned down our claims throughout the manuscript:

We have revised the title, abstract, conclusion and the relevant sections in the main text to redefine our framework as an "interpretable candidate drug prioritization and explanation framework" rather than a discovery platform.

We have repositioned the study as a hypothesis-generation tool designed to assist experts in prioritizing candidates for further experimental validation.

We have explicitly added a new "Limitations and Future Work" section (Section 5.6) in the Discussion chapter, where we acknowledge the lack of direct experimental validation and discuss the inherent constraints of the current datasets.

Comment 3: The interpretability framework is largely heuristic and is not itself validated...

Response: We sincerely appreciate the reviewer's rigorous standard for the empirical justification of our scoring framework and the biological relevance of the identified paths:

1.Empirical Justification via Ablation and Sensitivity Analysis.

We would like to clarify that, in response to the constructive feedback from the previous round of review, we had already incorporated a Parameter Sensitivity Analysis (Section 5.3) and an Ablation Study (Section 5.4) into the manuscript. As shown in Section 5.3 and Section 5.4, the manual weights and the confidence calibration factor λDFS are essential for maintaining the specificity of the ranking. Our results confirm that these heuristic constraints effectively suppress generic high-degree nodes, ensuring that the prioritized paths remain biologically relevant.

2.Clarification on Mechanistic Paths: Regarding the reviewer's concern about the specific nodes.

"Irritability": We would like to point out that the path involving "irritability" had already been filtered out in our previous revision of the manuscript (the second version) as it did not meet our refined specificity threshold.

"Bipolar Disorder" (Table 10, Path 7): We have retained this path because it represents a well-documented neuro-metabolic crosstalk mechanism for Topiramate. As supported by literature [45,46], modulating pathways related to bipolar disorder can indirectly influence energy metabolism and insulin sensitivity through the regulation of PVALB. We have further refined the discussion in Section 5.5 to clarify that this is a secondary regulatory mechanism, reflecting the complex systemic nature of antidiabetic therapeutics.

Comment 4: The manuscript contains a major internal inconsistency in the model comparison.

Response: We sincerely apologize for any confusion caused by the clerical inconsistency in the original version of our manuscript. We wish to clarify that in the current revised version, we have meticulously synchronized the results across the Abstract, Results, and Conclusion sections. We confirm that ComplEx consistently outperformed all other baseline models, and this conclusion is now accurately reflected throughout the entire paper.

Comment 5: There is insufficient rigor in the quantitative evaluation.

Response: We sincerely appreciate the reviewer's professional suggestion regarding the statistical rigor of our quantitative evaluation. To address this, we have systematically re-evaluated all four models using three different random seeds (42, 123, and 2026) to assess their stability.

Following your advice, we have updated Table 6 to report the primary link prediction metrics (MRR and Hits@K) as Mean ± Standard Deviation (SD). To manage the immense computational overhead of our 7.1-million-triple graph while strictly guaranteeing a fair comparison, these stability tests were conducted under identical parameters for 30 epochs. As shown in the updated table, the extremely low standard deviations (e.g., SD±0.003 for ComplEx Hits@10) empirically validate that the superior performance of the ComplEx model is statistically significant and not due to stochastic initialization.

Furthermore, to ensure absolute clarity regarding our methodology, we have refined Section 4.2 (Experimental Settings) and Section 5.1（Link Prediction Performance） to detail our two-stage evaluation strategy:We utilized the multi-seed tests (30 epochs) as rigorous statistical validation to establish model stability and relative ranking.Once the superiority of ComplEx was empirically proven, the final downstream candidate prioritization was performed using the fully converged ComplEx model (trained for 150 epochs) to maximize predictive sensitivity and biological accuracy.

We believe this approach fulfills both the rigorous statistical requirements of model comparison and the practical need for high-precision biological discovery.

Comment 6: Entity alignment and relation fusion need validation.

Response: We appreciate the reviewer’s constructive comments regarding the validation of our graph construction. Given the massive scale of the CMKG (over 7.1 million triples), exhaustive manual verification is unfeasible. Therefore, we adopted a stratified random sampling strategy to evaluate the most representative and cross-systemically integrated components of the graph.

1.Rationale and Methodology for Validation.

To ensure the high-fidelity integration of Traditional Chinese Medicine (TCM) and Modern Medicine（MM）, we selected Genes, Diseases, and Herbs as the target domains for validation. These entities serve as the primary semantic bridges in our graph and possess the widest coverage across the five source databases (Hetionet, SymMap, TCMBank, STRING, and TTD). By validating these high-degree nodes, we can robustly infer the reliability of the overall graph structure.

2.Quantitative Results of Entity Alignment.

We manually audited 100 randomly sampled entities (40 Genes, 30 Diseases, and 30 Herbs). By tracing each entity back to its original database identifiers and biological definitions, we observed an Alignment Accuracy of 98.0%. This confirms that our name-based reassignment protocol effectively maintains semantic consistency across heterogeneous sources.

3.Validation of Relation Fusion Logic.

To test the efficacy of our statistical thresholds (Jaccard ≥ 0.8 or Jaccard ≤ 0.2 , Overlap ≥ 0.6), we audited 60 triples from the merged and filtered subsets:

Accuracy of Merged Relations (93.3%): 28 out of 30 triples (primarily Anatomy-Gene pairs) were biologically rational. The minor errors (6.7%) highlighted a need for future Entity-Type Constraints to prevent compounds/diseases from inheriting tissue-specific predicates.

Accuracy of Evidence Filtering (86.7%): 26 out of 30 triples were correctly kept separate to preserve evidence granularity. We identified 4 false negatives where physical bindings (e.g., CALM1-PDE1A) were statistically underestimated by source databases.

Comment 7: Data and code availability are not yet adequate for reproducibility.

Response: We fully agree with the importance of reproducibility. We apologize for the confusion, which may have stemmed from the asynchronous nature of the previous revision cycle.While a repository was established in the previous revision, we have significantly updated it to include new resources generated in response to your specific feedback (Comments 5 and 6).The updated repository is available at:https://github.com/kunkunabc/T2DM-CrossMed-KG/tree/main.

Key additions to the repository include:

Statistical Rigor (Comment 5): Complete scripts and raw results for multi-seed training (seeds 42, 123, 2026) across all models to ensure performance stability.

Graph Validation (Comment 6): Stratified sampling scripts and the manual audit tables (98.0% entity accuracy and 93.3% relation rationality) providing the ground-truth verification requested.

We have updated the Data Availability Statement in the manuscript to include this link, ensuring all data and code are fully accessible for peer review and future research.

Comment 8: The writing requires substantial editing before publication.

Response: We apologize for the grammatical and stylistic errors. We have conducted a comprehensive linguistic proofreading of the entire manuscript. Specifically:

We have corrected the identified typos, such as changing "Traditional chinese medicine" to "Traditional Chinese Medicine," and "burnden" to "burden."

We have rectified inconsistent capitalization (e.g., "Mediacal Insurance") and punctuation errors (e.g., missing spaces after periods).

We have standardized the use of terminology and improved the sentence structures for better clarity and academic rigor.

In addition, we further checked other grammatical expressions and other issues throughout the text and highlighted them in the Revised Manuscript with Track Changes.

We hope that these revisions adequately address your concerns and make the manuscript suitable for publication.

Sincerely,

Zekun Zhou

Corresponding Author:Shuo Yang

Affiliation:Institute of Information on Traditional Chinese Medicine, China Academy of Chinese Medical Sciences

---

## [Editor Report · Decision Letter 2]

23 Apr 2026

Interpretable Candidate Drug Prioritization and Explanation Framework Across-Medical Knowledge Graphs Based on Graph Embedding Models: A Case Study of Type 2 Diabetes

PONE-D-25-64588R2

Dear Dr. Zhou,

We’re pleased to inform you that your manuscript has been judged scientifically suitable for publication and will be formally accepted for publication once it meets all outstanding technical requirements.

Kind regards,

Zeheng Wang

Academic Editor

PLOS One
---

## [Editor Report · Acceptance letter]

PONE-D-25-64588R2

PLOS One

Dear Dr. Zhou,

I'm pleased to inform you that your manuscript has been deemed suitable for publication in PLOS One. Congratulations! Your manuscript is now being handed over to our production team.

Kind regards,

on behalf of

Dr. Zeheng Wang

Academic Editor

PLOS One